# Impact of climate change on snow avalanche activity in the Swiss Alps

Stephanie Mayer[1], Martin Hendrick[1], Adrien Michel[2], Bettina Richter[1], Jürg Schweizer[1], Heini Wernli[3], and Alec van Herwijnen[1]

[1]WSL Institute for Snow and Avalanche Research SLF, Davos, Switzerland
[2]Federal Office of Meteorology and Climatology MeteoSwiss, Geneva, Switzerland
[3]Institute for Atmospheric and Climate Science, ETH Zurich, Zurich, Switzerland

**Correspondence:** Stephanie Mayer (stephanie.mayer@slf.ch)

**Abstract.** The cryosphere in mountain regions is rapidly transforming due to climate change, yet the impact of these changes on snow avalanche activity remains uncertain. Here, we use a snow cover model driven by downscaled climate projections to evaluate future alterations in dry- and wet-snow avalanche occurrences throughout the 21st century in the Swiss Alps. We assess avalanche activity by employing machine learning models trained with observed records of avalanches. Our findings indicate an overall decline in the occurrence of dry-snow avalanches during the months December to May that is partially compensated by an increase in wet-snow avalanche activity. Depending on elevation and the emission scenario considered, we anticipate a net reduction in the total avalanche activity ranging from under 10% to as much as 60% by the end of the century compared to 45-75 avalanche days/year at the beginning of the century. Projections further reveal a shift of wet-snow avalanche activity to earlier winter months. Analysis of changes in prominent snow grain types offers a coherent explanation of projected changes beyond a mere decrease in snow depth and snow cover duration. Overall, our study quantifies for the first time the significant influence of climate change on snow avalanche activity in the Swiss Alps and may serve as a benchmark for further mountain regions with similar avalanche climates.

## 1 Introduction

Snow avalanches pose a significant natural hazard in seasonally snow-covered mountain regions around the globe, threatening human lives and infrastructure. Ongoing climate change is substantially changing the duration, extent and thickness of the seasonal snow cover, and consequently may affect the frequency, magnitude, and spatial occurrence of snow avalanches (Hock et al., 2019). Indeed, warmer temperatures lead to a shift from solid to liquid precipitation and an earlier snow-melt onset. Numerous studies have quantified the resulting decline of the seasonal snow cover (e.g. Mote et al., 2018; Smith and Bookhagen, 2018; Matiu et al., 2021), and climate projections suggest a continuation of this trend (e.g. Terzago et al., 2014; Fyfe et al., 2017; Verfaillie et al., 2018; Kotlarski et al., 2023). Yet, the impact of climate change on snow avalanches remains unclear, as snow avalanches are triggered by the complex interaction of weather and terrain with the snowpack, rather than just by the amount of snow on the ground (Schweizer et al., 2003). Mere knowledge about changes in duration and depth of the seasonal snow cover is therefore not sufficient to determine the effect of climate change on snow avalanche occurrence.

Throughout a winter season, distinct snow layers are deposited and continuously evolve under the influence of environmental factors such as precipitation, temperature, wind, and radiation. Snow stratigraphy, i.e. the vertical layering of the snowpack, is a key factor in the formation of snow avalanches. Dry-snow slab avalanches, the most destructive avalanches, release due to the failure of a mechanically weak snow layer buried below a cohesive slab (e.g. Bobillier et al., 2021; Bergfeld et al., 2023). Most dry-snow slab avalanches release naturally during or shortly after a snowstorm, but artificial triggering, for instance by skiers, is also possible. Wet-snow avalanches, on the other hand, form when liquid water from melting or rain-on-snow events infiltrates the snowpack and reduces the strength of existing weak layers (Baggi and Schweizer, 2009; Mitterer and Schweizer, 2013). Due to the differences in underlying physical processes, it is essential to distinguish between wet- and dry-snow avalanches when assessing the influence of climate change.

Few historical avalanche data series are of sufficient length, completeness and homogeneity to detect climate-related trends (Laternser and Schneebeli, 2002; García-Hernández et al., 2017; Favillier et al., 2023). Available studies indicate that rising temperatures have led to a decrease in the number and size of avalanches (Teich et al., 2012; Eckert et al., 2013; Giacona et al., 2021; Peitzsch et al., 2021), especially at lower elevations, and to an increase in the proportion of wet-snow avalanches (Pielmeier et al., 2013; Naaim et al., 2016). The influence of climate change on snowpack stability at higher elevations, where temperature increases have a weaker effect on the snow-rain partitioning, remains largely unclear. Some studies suggested an increase in dry-snow avalanche activity at higher elevations due to a warming-related intensification of heavy snowfall events (Lavigne et al., 2015), while others attributed an observed increase in avalanche activity to an enhanced likelihood of wet-snow conditions (Ballesteros-Cánovas et al., 2018). However, these studies either had limited data quality or were rather local in scope, making it difficult to predict future changes at a larger scale including higher elevations where most avalanches release.

To evaluate changes in snow avalanche activity for future decades, climate projections can be coupled with physics-based snow cover models that resolve the temporal evolution of snow stratigraphy required to assess snow instability. The few existing assessments of future snow stability based on simulated snow cover data (Lazar and Williams, 2008; Castebrunet et al., 2014; Katsuyama et al., 2022) relied either on bulk snowpack or meteorological parameters without accounting for the detailed snow stratigraphy, or employed a stability index that is a poor predictor of avalanche activity (Jamieson et al., 2007; Mayer et al., 2023). Thus, there is insufficient understanding of potential changes in avalanche activity within the current century and whether avalanche risk mitigation measures need to be adapted (Eckert et al., 2024).

Our goal is therefore to quantify how and why natural avalanche activity will change for different elevations in the Swiss Alps throughout the 21$^{st}$ century. We use machine learning models to classify dry- and wet-snow avalanche days based on downscaled climate change scenarios and simulated snow stratigraphy. Specifically, we investigate future changes in the occurrence of natural avalanches at mid- to high elevations using the CH2018 climate change scenarios (CH2018, 2018) and considering three different emission scenarios (Representative Concentration Pathways, RCP2.6, RCP4.5 and RCP8.5). We apply statistical methods to spatially transfer climate projections from eight members of the CH2018 ensemble to seven automatic weather stations (AWS) located close to typical avalanche starting zones in the Swiss Alps at elevations ranging from 1800 to 2900 m a.s.l. The downscaled meteorological variables serve as input for the physics-based snow cover model SNOWPACK (Bartelt and Lehning, 2002; Lehning et al., 2002a, b), yielding one-dimensional simulations of snow stratigraphy. To identify dry- and

wet-snow avalanche days (AvDs), we employ two recently developed classification models (Mayer et al., 2023; Hendrick et al., 2023). Based on the resulting transient time series of future avalanche activity, we finally examine changes in the number of avalanche days per winter season, explore changes in the seasonal fluctuations of avalanche activity, and investigate causes of the predicted changes using the simulated snow stratigraphy.

## 2   Data and Methods

To project changes in dry- and wet-snow avalanche activity throughout the 21$^{st}$ century, we established a model chain to (1) downscale climate projections, (2) simulate snow stratigraphy, and (3) classify dry- and wet-snow avalanche activity (AvD vs. non-AvD) at seven sites in the Swiss Alps on a daily basis.

### 2.1   Selection of sites

We aimed to project changes in avalanche activity for an elevation range where, even under strong emissions scenarios, a continuous snow cover over an extended time period is still likely to exist by the end of the century. In mountain regions, the near-surface zero degree line roughly demarcates the regions where precipitation falls predominantly as either snow or rain (Hock et al., 2019). Under strong emissions (RCP8.5), the winter zero-degree line averaged over the country of Switzerland is projected to increase in elevation from today's level at 850 m a.s.l. to up to 1800 m a.s.l. by the end of the century (CH2018, 2018). We therefore selected seven sites at an elevation range above 1800 m a.s.l. where the majority of potential avalanche release areas are located in the Swiss Alps (Bründl et al., 2019). The location of those sites corresponded to the location of AWS of the IMIS network used in operational avalanche forecasting in Switzerland (Lehning et al., 1999; SLF, 2022). The CH2018 climate scenarios are however only available for AWS from the SwissMetNet network of the Federal Office of Meteorology and Climatology MeteoSwiss, which are usually located at lower elevations (< 1800 m a.s.l.). To enable a spatial statistical transfer of these climate projections to the higher-elevation IMIS stations, we selected the AWS from the IMIS network according to the following criteria: (1) sufficiently long time series ($\geq$ 20 years) of continuously measured data, (2) not more than two consecutive days of missing snow depth measurement, and (3) availability of a SwissMetNet AWS included in the CH2018 dataset situated less than 30 km apart in the same climate regime (northern vs. southern flank of the Alps). The geographical positions of the IMIS stations and their associated SwissMetNet stations are depicted in Fig. 1.

The selected seven sites cover different snow climates (Schweizer et al., 2024). According to Mock and Birkeland (2000), the snow climate is classified as maritime and partly transitional for three stations (VDS2, SCH2, ORT2), transitional to maritime for two stations (ZER2, KLO2), and transitional to continental for the remaining two stations (ARO2, WFJ2).

### 2.2   Climate projections CH2018

Climate projections used in this study are based on the CH2018 climate change scenarios (CH2018, 2018) which are available for three Representative Emission Pathways (RCPs) each named according to the level of radiative forcing (in $\mathrm{W\,m^{-2}}$) projected to be reached by the end of the century (Stocker et al., 2013): Scenario RCP2.6, corresponds to strong reductions in

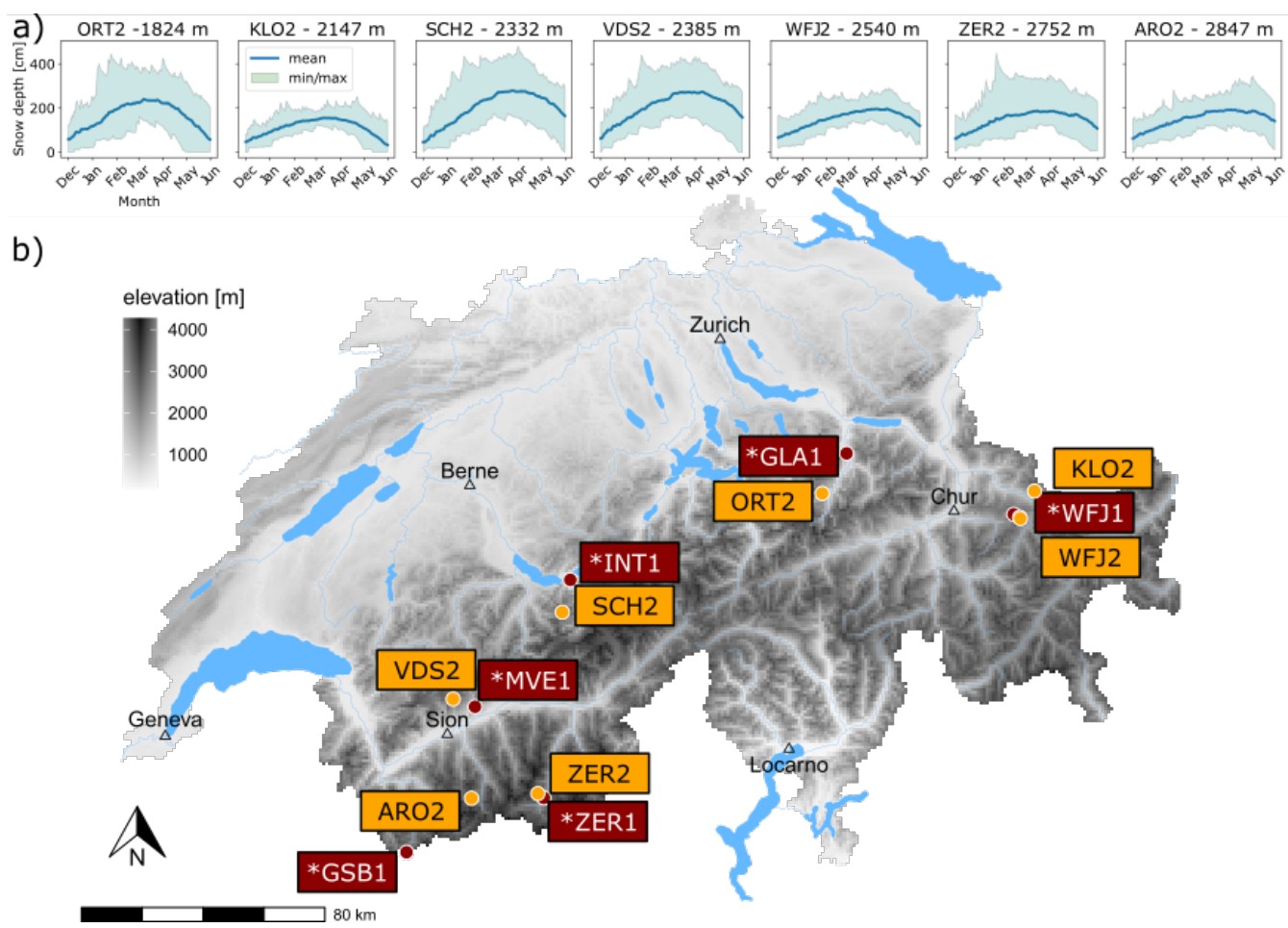

**Figure 1.** (a) Mean (blue line) and spread (shaded area, determined by the spread between daily minima and maxima) in measured snow depth from 2001 to 2022 for the seven IMIS AWS. (b) Location of the IMIS AWS (yellow dots) and the corresponding SwissMetNet AWS (dark red).

global emissions likely limiting global warming to below 2 °C by 2100. In the intermediate scenario RCP4.5, greenhouse gas emissions peak around mid-century and then decline due to moderate mitigation efforts. The high-emission, "worst-case" scenario RCP 8.5 is characterized by a lack of significant mitigation measures and continuous growth in atmospheric greenhouse gas concentrations, without considering potential fossil fuel limitations.

Based on the set of RCPs, projections of changes in the climate system were simulated using an ensemble of General Circulation Models (GCMs), and the resulting outputs were then used as boundary conditions for a Regional Climate Model (RCM) ensemble over Europe, as part of the EURO-CORDEX experiment (Jacob et al., 2014). The CH2018 climate scenarios were finally obtained from a statistical downscaling of the EURO-CORDEX ensemble of coarser RCM simulations to AWS using quantile mapping (QM; Gudmundsson et al., 2012; Rajczak et al., 2016). QM is an established statistical method used to transform the distribution of modelled data so that its statistical properties align more closely with observed data. When applied in a climate modelling context, QM involves building a transfer function that relates quantiles from the modelled distribution of a meteorological variable to the corresponding quantiles in the observed distribution for a given reference period. The transfer function is then applied to bias-correct a transient time series of the simulated variable. QM is thus primarily a bias-correction method but can implicitly serve as a downscaling method if the observed reference data refer to a higher spatial resolution than the climate model (Feigenwinter et al., 2018).

In the context of the CH2018 scenarios, QM implies that when the daily precipitation sum on a given day in a specific RCM grid cell equals the 70th percentile of the reference period's climate simulations, the daily precipitation sum at a corresponding AWS is set to the 70th percentile of the observed daily precipitation at that station during the reference period. In principle, this implies that the downscaling preserves realistic weather patterns and the downscaled data capture realistic storm events that are present in the RCM simulations if those events are represented in the reference period. However, QM assumes that the relation between conditions in the grid cell and the station is stationary, which might not hold true under changing climatic conditions. Moreover, the QM transfer function is subject to uncertainties due to the limited number of years considered for the calibration, in particular for high and low quantiles. Furthermore, the extremes that have not been observed in the reference period are mapped using a constant extrapolation of the biases for the 1st and 99th percentiles (CH2018, 2018). These limitations should be particularly considered when interpreting results related to extreme events. The CH2018 scenarios are available at a daily resolution for seven variables (daily mean, minimum and maximum air temperature, precipitation, relative humidity, wind speed and incoming shortwave radiation) for a set of AWS from the SwissMetNet network and cover the complete time period 1981 - 2099.

## 2.3 Downscaling of climate projections

To spatially transfer the CH2018 scenarios for the SwissMetNet AWS to the corresponding seven IMIS AWS situated at the elevation of potential avalanche starting zones, we used a combination of univariate QM (see Sect. 2.2) and multivariate QM (MQM; Cannon, 2018). MQM is a multivariate generalization of QM where the transfer function is based on the multidimensional distribution of different meteorological variables. The MQM approach allowed for bias correction while preserving more realistic inter-variable relationships, providing a distinct advantage over other statistical methods for bias correction. Moreover,

it was necessary to downscale the climate scenarios to an hourly resolution, as required to force the SNOWPACK model. In
more detail, we conducted the following steps:

1. Transfer of daily climate change scenarios: The CH2018 climate scenarios are based on a spatial transfer of EURO-
   CORDEX data to SwissMetNet AWS using univariate QM (see Sect. 2.2). However, univariate QM partially removes
   correlations between variables (Michel et al., 2021), and EURO-CORDEX data have an unrealistic relationship between
   precipitation and temperature, with more precipitation on warm days than on cold days as compared to measurements
(Meyer et al., 2019). Consequently, solid precipitation, and thus snow depth, is underestimated when forcing snow mod-
   els with CH2018 data. We therefore used MQM (Cannon, 2018) to restore inter-variable correlations in climate change
   scenarios based on those observed in the historical time series (the historical IMIS time series), thereby also removing
   the precipitation bias in the underlying EURO-CORDEX data. Nevertheless, the MQM of Cannon (2018) uses the same
   bias correction throughout the year without any seasonality. We therefore first corrected for seasonal bias by applying a
univariate QM for each day of the year (as in CH2018, 2018) using the method of Kotlarski (2019), and subsequently
   applied the MQM of Cannon (2018). Using this combination of QM and MQM improved the simulated snow depth over
   the historical period. Due to the restoration of inter-variable correlations with MQM, we expect that our approach better
   captures storm characteristics driving avalanche-relevant snow stratigraphy compared to other statistical bias correction
   methods such as the delta change method (Michel et al., 2021). The QM was applied to the following variables: daily
mean, minimum and maximum air temperature, precipitation, relative humidity, wind speed and incoming shortwave
   radiation.

2. Downscaling climate change scenarios to hourly values: The climate change scenarios obtained through QM inherit the
   daily resolution of the CH2018 scenarios. We therefore used the MEteoroLOgical observation time series DISaggrega-
   tion Tool (MELODIST) library (Förster et al., 2016) to reconstruct the daily cycle based on the seven variables obtained
in the previous step. This software consists of disaggregation functions to reconstruct daily cycles of meteorological
   variables. Multiple functions, purely static or based on statistics of past observations at the AWS of interest are available.
   The parameters used for the MELODIST library are detailed in Appendix A (Table A2). Finally, hourly resolution cli-
   mate change scenarios were obtained for precipitation, air temperature, incoming shortwave radiation, wind speed and
   relative humidity.

The downscaling process was implemented individually for each of the seven IMIS AWS, using the historical time series
for the IMIS AWS (see Appendix B) and the CH2018 climate simulations for the corresponding nearby Swissmetnet AWS and
the same historical time period. For each IMIS AWS, the downscaling scheme was applied to eight climate model chains of
the CH2018 ensemble which are listed in Appendix A (Table A1). Selection criteria for the model chains included availability
across all three RCP scenarios (RCP2.6, RCP4.5, and RCP8.5) and inclusion of all variables necessary to drive SNOWPACK.
The downscaled climate projections for the IMIS AWS cover the time period 1990 - 2099. The same downscaling scheme was
applied by Ortner et al. (2023) to investigate the impact of climate change on avalanche runout.

## 2.4 SNOWPACK simulations

To simulate flat-field snow stratigraphy at the sites of the seven AWS, we used the physics-based model SNOWPACK (Bartelt and Lehning, 2002; Lehning et al., 2002a, b). Based on meteorological input data, SNOWPACK simulates the one-dimensional vertical snow stratigraphy over time with snow layer thicknesses on the order of centimeters. Distinct layers are described with microstructural properties, such as grain and bond size (order of mm), and bulk properties, such as density and liquid water content. Snow grain types are classified according to the microstructural properties (Lehning et al., 2002b; Fierz et al., 2009). Several validation campaigns have demonstrated that modeled and observed snow stratigraphy generally agree well, in particular with respect to critical weak layers (e.g. Lehning et al., 2001; Horton et al., 2014; Richter et al., 2019; Calonne et al., 2020).

Climate projections of snow stratigraphy from 1 September 1990 to 31 May 2099 were obtained by driving SNOWPACK with the hourly downscaled climate scenarios. Simulations were conducted using a computation step size of 15 minutes. To model the soil heat flux at the bottom of the 3 m deep soil column below the snowpack, we chose a standard constant value of $0.06 \, \mathrm{W/m^2}$ (Davies and Davies, 2010). The full energy balance at the snow-atmosphere boundary was calculated (Neumann boundary condition). Incoming longwave radiation, which was not included in the climate downscalings, was obtained using the parametrization of Carmona et al. (2014). The flow of liquid water through the snow cover was simulated based on a bucket scheme approach (Bartelt and Lehning, 2002).

For validation purposes, we also performed SNOWPACK simulations using meteorological measurements from the seven AWSs (Sect. 3). In this case, SNOWPACK settings were slightly different from those used for the climate projections, as a different set of meteorological variables was available. Specifically, simulations were driven with measured snow depth rather than with precipitation. Furthermore, we imposed the measured snow surface temperature as upper Dirichlet boundary condition at the snow-atmosphere boundary. Whenever the snow surface temperature exceeded -1 °C, upper boundary conditions switched to Neumann-type, preventing an underestimation of energy input during ablation periods. All further settings were identical to those used for the climate projections.

For the analysis of simulated snow stratigraphy, we grouped the simulated primary snow grain types into four classes to combine grain types with similar microstructural characteristics and similar significance with respect to avalanche formation (Table 1). Individual grain types are described in Fierz et al. (2009).

## 2.5 Assessment of avalanche activity

To assess dry-snow avalanche probability in the vicinity of the AWS, we used a model that evaluates information on potential weak layers in the simulated snow stratigraphy in combination with the simulated new snow amounts (Model "combi" in Mayer et al., 2023). Potential weak layers were identified using a random forest model providing a probability of instability for every layer in the simulated snow stratigraphy (Mayer et al., 2022). The daily maximum of the probability of instability as well as the three-day sum of new snow were then combined into a probability of natural avalanche activity based on statistical models that were fit to observed avalanche activity data (Mayer et al., 2023). This dry-snow avalanche model was trained and

**Table 1.** Definition of grain type classes.

| Grain type class | members |
| --- | --- |
| Precipitation particles | Precipitation particles |
| | Decomposing and fragmented precipitation particles |
| Rounded grains | Rounded grains |
| Persistent grain types | Faceted crystals |
| | Depth hoar |
| | Surface hoar |
| | Rounding faceted particles |
| Melt forms | Melt forms |
| | Melt-freeze crust |
| | Ice formations |

extensively validated using observations of snow stratigraphy, snow instability and avalanches (for more details, see Mayer et al., 2023, 2022).

Wet-snow avalanche probability in the vicinity of an AWS was computed with the random forest model described in Hendrick et al. (2023). The model was trained to differentiate between days with or without wet-snow avalanche activity based on a data set of wet-snow avalanche observations spanning 20 years. Its input features include meteorological variables, e.g. the daily incoming short-wave radiation, as well as variables related to simulated snow stratigraphy such as the maximum of the liquid water content among all snow layers. The wet-snow model was validated in both nowcast (based on measured variables) and forecast mode (based on variables computed from a numerical weather prediction model, Hendrick et al., 2023).

A day was classified as a dry-snow AvD or wet-snow AvD if the respective model indicated a probability of avalanche occurrence greater than 0.5 and the simulated snow depth was at least 40 cm. For the winter seasons from 1990/1991 through 2098/2099, every day within the period December to May (DJFMAM) was categorized as either an AvD or non-AvD with respect to dry- and wet-snow conditions based on the simulated meteorological and snow stratigraphy data. This classification was performed for each of the seven AWS, and for all 24 possible combinations of climate models (N=8) and RCP scenarios (N=3).

## 2.6 Analysis of projected changes

To evaluate how and why avalanche activity will change throughout the 21$^{st}$ century we analyzed

- changes in the number of simulated dry- and wet-snow AvDs within the period DJFMAM

- shifts in the timing of dry- and wet-snow avalanche occurrence using monthly averages

- changes in the proportion of snow grain types within the the DJF snowpack.

All analyses were based on averages over 30 year periods which allows capturing the climatological influence. As a reference
period (REF) we used the 30 year period spanning the winter seasons 1990/1991-2019/2020. We further defined the end-of-the-
century period (EOC) as the winter seasons 2069/2070-2098/2099. To assess the statistical significance of projected changes,
we used Mann-Kendall (MK) test statistics (Mann, 1945; Kendall, 1975).

## 3  Validation

To investigate the accuracy of our model chain, we validated the downscaled climate scenarios as well as the simulated
avalanche activity for the period where continuous meteorological measurements were available for the seven IMIS AWS
(winters 2000/2001-2021/2022). We compared the following variables for the months DJFMAM averaged over the period
2000/2001-2021/2022:

- Seasonal number of AvDs, comparing predictions of the avalanche activity models based on SNOWPACK simulations
  either forced with the downscaled climate models or with the measurements from the IMIS AWS.

- Air temperature, comparing downscaled climate scenarios with measurements from the IMIS AWS.

- Snow depth, comparing SNOWPACK simulations forced with downscaled climate models with measurements from
  IMIS AWS.

- Three-day sum of new snow height provided by SNOWPACK (i.e. sum of three consecutive values of the height of new
  snow fallen within 24 hours), comparing simulations either forced with downscaled climate models or with measure-
  ments from the IMIS AWS. The AWS measurements provide snow depth only and do not measure new snow height.
  Hence, for this case, SNOWACK was driven with snow depth measurements, and new snow height was calculated tak-
  ing into account settlement of the snowpack. For the simulations driven with the downscaled climate model output,
  SNOWPACK was forced with precipitation and the height of new snow was again computed considering settlement.

The validation of the avalanche activity models is presented in Fig. 2 for all emission scenarios. Results of the validation of
the meteorological variables are presented in Appendix C (Fig. C1 on a monthly basis for emission scenario RCP8.5 and Fig.
C2d,h,l on a seasonal basis for all RCP scenarios). The results for the seasonal number of AvDs, snow depth, and three-day sum
of new snow height account for the uncertainties in the reconstructed historical IMIS time series (precipitation and incoming
shortwave radiation), uncertainties induced by the use of climate change scenarios (i.e. the part not corrected by the QM), and
uncertainties induced by SNOWPACK in deriving snow depth and three-day sum of new snow height.

To validate the complete model chain, including both dry- and wet-snow avalanche activity models, we calculated the sea-
sonal number of avalanche days (AvDs) using SNOWPACK simulations. These simulations were forced with either downscaled
climate models or AWS measurements, and we then computed the relative difference between the resulting seasonal numbers
of AvDs (Fig. 2). For the two lowest stations, the simulations forced with climate models resulted in an over-prediction of
dry-snow AvDs (about 20% for ORT2 and KLO2 in Fig. 2a), while for the other stations differences were smaller, typically an

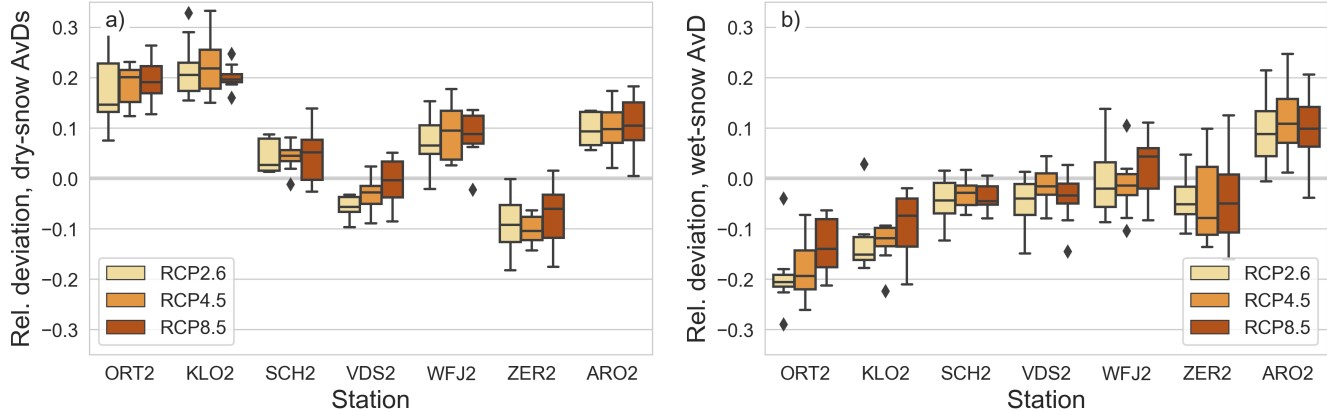

**Figure 2.** Relative differences between the simulated average seasonal (DJFMAM) number of AvDs based on SNOWPACK simulations forced with the downscaled climate models for the 2000/2001-2021/2022 period and the average seasonal number of AvDs based on SNOW-PACK simulations forced with measurements from the same time period. Results are shown for relative differences regarding dry-snow (a) and wet-snow (b) AvDs. Colors indicate the different RCP scenarios. The box and whiskers show the variability within the eight ensemble members, where the line indicates the median, the box extends from the first to the third quartile, the whiskers extend to 1.5 the interquartile range beyond the box limit, and the black diamonds show outliers. The stations are arranged in order of increasing elevation, from the lowest (ORT2, 1824 m a.s.l.) to the highest (ARO2, 2847 m a.s.l.).

over- or under-prediction on the order of 10% or less. Regarding the seasonal number of wet-snow AvDs, there was a rather clear trend with elevation, with an under-prediction of about -20% for the lowest station, and an over-prediction of about 10% for the highest station. Considering the natural daily variability in avalanche activity and the discrepancies in the input data (Fig. C1), the relative differences in the predicted numbers of dry- and wet-snow AvDs are still rather low. Note that we could not validate the AvD predictions with visual avalanche observations, since observational data are not consistently available

for the 20-year period we considered. The avalanche prediction models, on the other hand, were thoroughly validated with quality-controlled sets of avalanche observations (Hendrick et al., 2023; Mayer et al., 2023).

For air temperature, snow depth and three-day sum of new snow height, we first computed the day-of-year average for the variable of interest considering 22 values from the climate simulations spanning the period 2000/2001-2021/2022. For each of the climate models, we then compared these values to the corresponding values based on the observed data (Appendix C, Fig.

C1). The bias in air temperature was negligible and the RMSE was about 1 °C, well within daily natural variability. Differences in snow depth were largest for the station at the lowest elevation (ORT2) and generally decreased with elevation. Overall, simulated values of snow depth were in good agreement with measurements and well within typical variations observed in mountainous terrain (e.g. Helbig and van Herwijnen, 2017), and for five out of seven stations the bias in snow depth was below 10%. Finally, three-day sums of new snow height simulated either using downscaled climate model data or using IMIS

measurements compared well for all stations, with a slight tendency to under-predict the amount of new snow in April and May.

## 4 Results

### 4.1 Response of avalanche activity to warming

By the end of the 21st century, our models project substantial changes in avalanche activity during the months DJFMAM. Changes in dry-snow, wet-snow, and total avalanche activity relative to the 30-year REF period are depicted using 30-year moving means of the projected seasonal number of AvDs up until the EOC (Fig. 3). Regarding dry-snow avalanche activity (Fig. 3b-e), the multi-model mean indicates a decrease across all elevations, with the most significant reductions projected towards the EOC for all RCP scenarios. The largest declines, up to 65%, are observed under the RCP8.5 scenario. While the eight ensemble members show considerable variation, with some indicating a potential increase of up to 10% in dry-snow AvDs, particularly during the first half of the century, the majority of members predict a statistically significant (MK test) decline in dry-snow avalanche activity by the EOC for all emission scenarios.

Conversely, wet-snow avalanche activity (Fig. 3g-j) during the months DJFMAM will increase for stations above 2300 m a.s.l., and decrease for stations at lower elevations. Under RCP2.6 and RCP4.5, the number of wet-snow AvDs projected by the ensemble mean levels off around mid-century at relative increases of 5-20% for stations above 2300 m a.s.l. and at relative decreases of 15-20% for lower stations compared to the REF period. Under the RCP8.5 scenario, stations above 2300 m a.s.l. are projected to experience a peak in the number of wet-snow AvDs during the second half of the century, with a relative increase of 10-30%, followed by a subsequent decrease. For the two stations below 2300 m a.s.l., the decrease in wet-snow avalanche activity is most pronounced towards the EOC and in particular for the RCP8.5 scenario, with a decrease by 50-60%. Uncertainty within the ensemble is large for all stations, with absolute differences up to 55% between individual model chains. For most combinations of climate models and RCP scenarios, increasing trends for stations above 2300 m a.s.l. up to the year 2070, and decreasing trends for the lower stations up to the EOC were statistically significant (MK test). Interestingly, the decrease in dry-snow avalanche activity is mostly offset by the increase in wet-snow avalanche activity, resulting in decreases of less than 10% in the total number of AvDs per DJFMAM for most stations, except for elevations below 2300 m a.s.l. or under the RCP8.5 scenario in the latter half of the century (Fig. 3l-o). Furthermore, we find a consistent trend with elevation, with more pronounced changes in projected avalanche activity occurring at lower elevations, particularly for wet-snow avalanches (Fig. 3e,j,o).

### 4.2 Changes in seasonality

To investigate changes in avalanche activity over the course of a winter season, we compared monthly averages (30-day moving sums) of AvDs between the REF and EOC periods (Fig. 4 for RCP8.5 and Appendix D, Fig. D1 for all RCPs). Seasonality in dry-snow avalanche activity remains mostly unchanged. Indeed, for both the REF and EOC periods, dry-snow avalanche activity generally peaks in January and February and decreases thereafter. Yet, the number of AvDs is much lower at the EOC (negative anomaly for all stations in Fig. 4c). The seasonal distribution of wet-snow AvDs, however, is projected to change. For the REF period, wet-snow avalanche activity between December and February is mostly close to zero and rapidly increases in March, peaking towards the end of the season (Fig. 4d). At the EOC, however, the increase in wet-snow avalanche

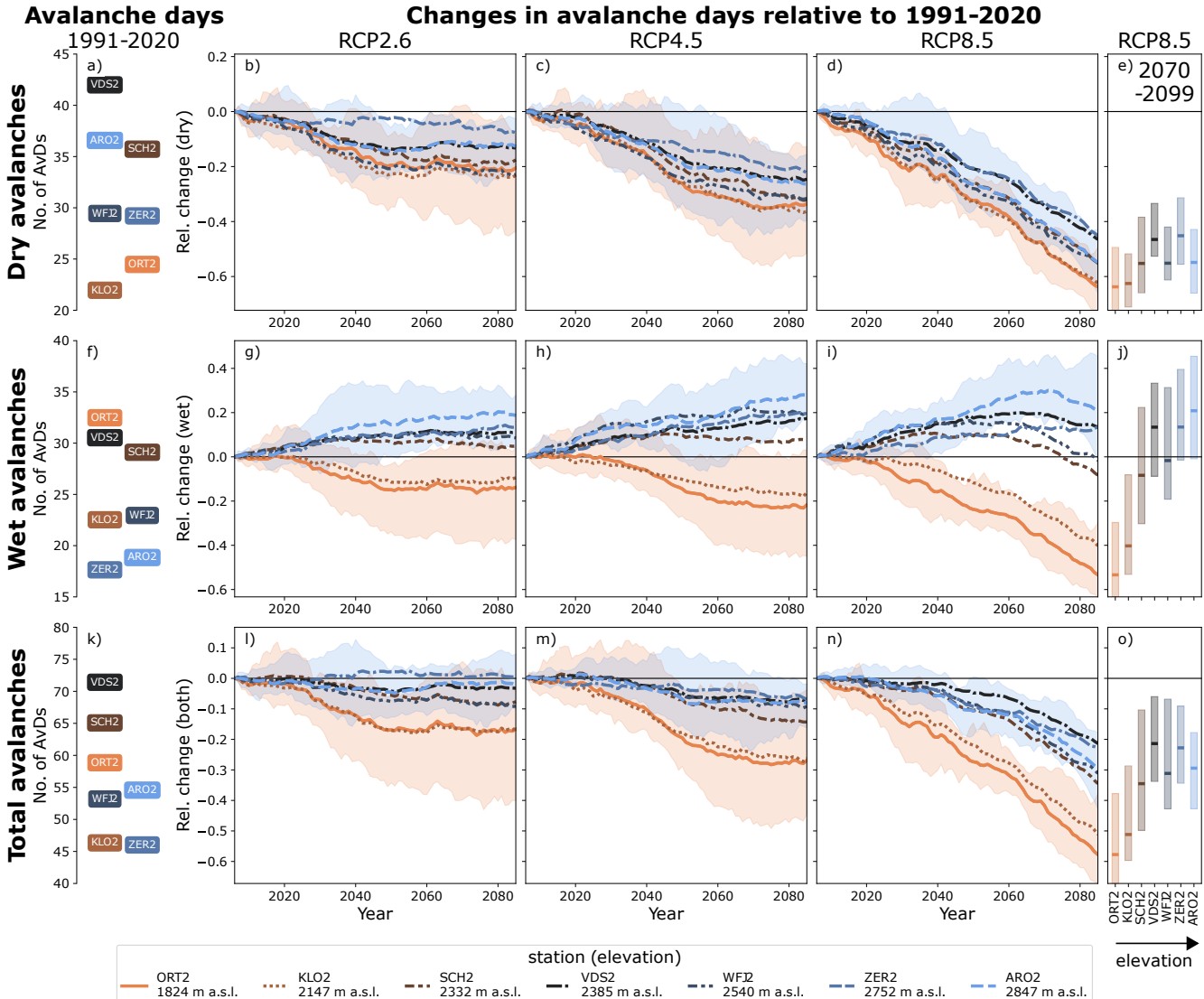

**Figure 3.** Projected changes in dry-snow (b-e), wet-snow (g-j), and total avalanche activity (l-o) for the 21st century (centered 30-year moving average relative to the reference period of 1991-2020) under three different emission scenarios (RCP2.6, RCP4.5, and RCP8.5). Lines show the multi-model mean of the seasonal (DJFMAM) number of AvDs for seven AWS across the Swiss Alps. Shaded areas indicate the spread of the eight climate ensemble members for the highest and lowest stations. Station identification is facilitated by colors and line styles specified in the bottom legend. The left panels (a,f,k) display the mean seasonal number of AvDs during the reference period for each station, the right panels (e,j,o) show the mean (line) and spread (colored bars) under RCP8.5 for the period of 2070-2099.

activity is more gradual throughout the season, resulting in a higher frequency of avalanches from December to March (positive anomaly for all stations except ORT2 in Fig. 4f). In April and May, though, there is an overall decrease in wet-snow avalanche

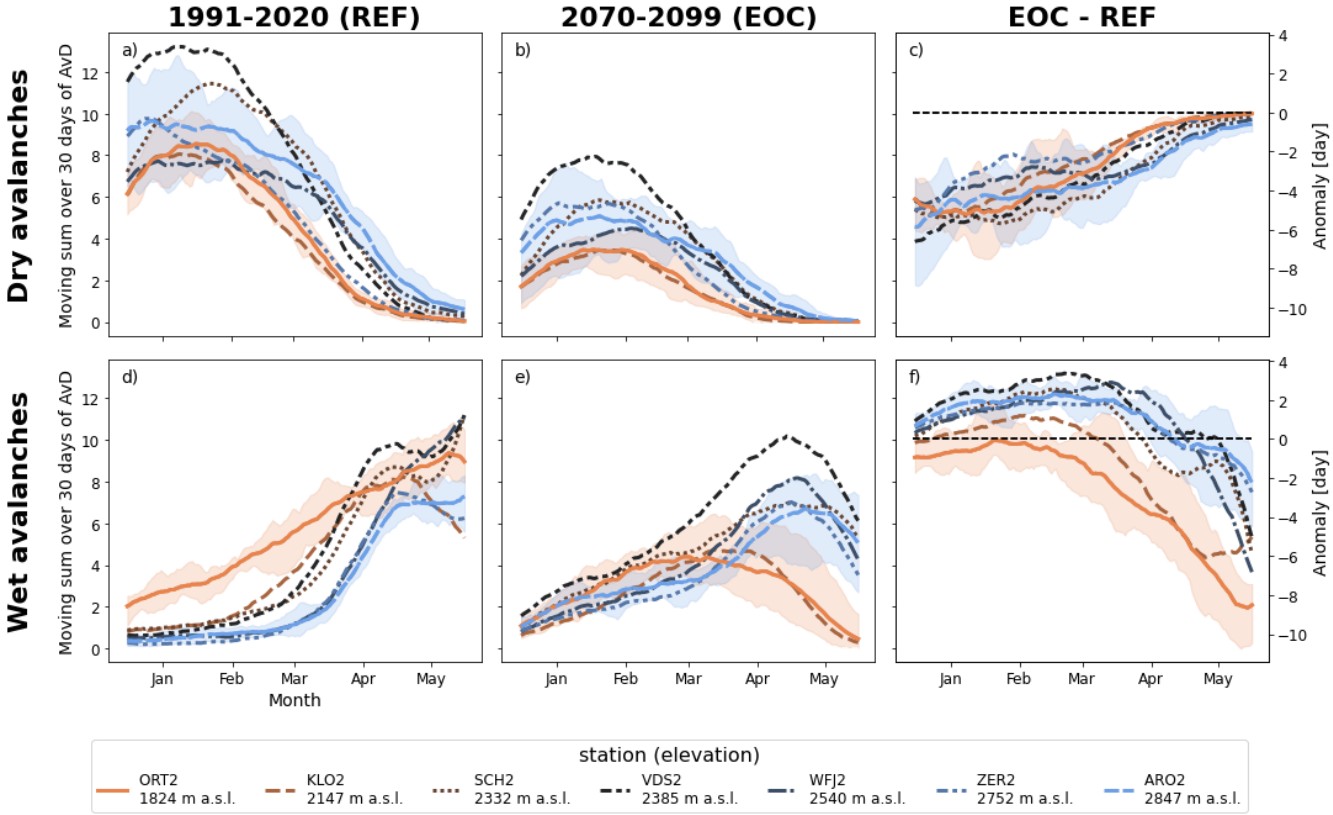

**Figure 4.** 30-day moving sum of AvDs for dry- (a,b) and wet-snow (d,e) conditions under RCP8.5 for the 1991-2020 (REF) and 2070-2099 (EOC) periods, as well as the resulting anomaly between EOC and REF (c,f). Lines indicate the multi-model mean over the considered period and shaded areas show the spread of the eight ensemble members for the highest and lowest stations.

activity compared to REF (Fig. 4f) and wet-snow avalanche activity will thus peak earlier in the season. Overall, these seasonal changes result in a decrease in the number of wet-snow AvDs for the two lowest stations (ORT2 and KLO2) and an increase for the two highest stations (ZER2 and ARO2), which is consistent across most models (Fig. 3g-i).

### 295 4.3 Causes of projected changes

Changes in avalanche activity can be attributed to both weather and snowpack factors. For instance, with higher temperatures there is a decrease in snow cover duration and snowfall amounts (Appendix C, Fig. C2), which results in fewer potential dry-snow AvDs. However, the relative decrease in dry-snow AvDs towards the EOC is notably larger (by more than 20% for most stations above 2300 m a.s.l.) than the relative decrease in days with sufficient snow depth and snowfall for natural

avalanche release (Appendix D, Fig. D2). Furthermore, the projected increase in wet-snow avalanche activity might also seem counter-intuitive with respect to the decrease in snow depth. These discrepancies can be attributed to projected changes in snow stratigraphy.

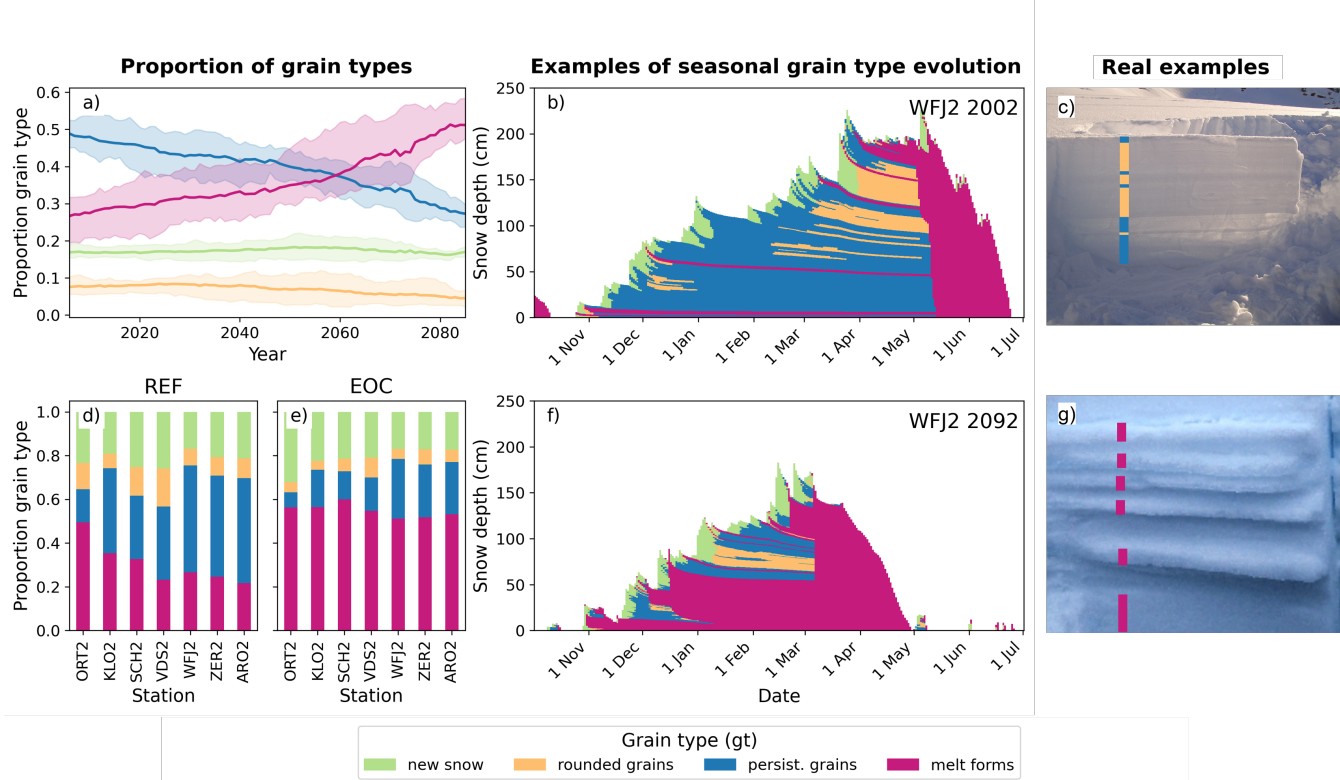

**Figure 5. a,d,e**: Average proportion of different grain type classes (indicated by colors) within the DJF snowpack under RCP8.5 for the WFJ2 site (2540 m a.s.l.) over the period 2006-2085 (centered 30-year moving average) (a) and for all stations averaged over the periods 1991-2020 (d) and 2070-2099 (e). Lines in (a) indicate the multi-model mean of the moving 30-year average proportion of the respective grain type and shaded areas show climate model spread. Bars in (d) and (e) refer to the multi-model mean. Stations in (d) and (e) are arranged in order of increasing elevation, from the lowest (ORT2, 1824 m a.s.l.) to the highest (ARO2, 2847 m a.s.l.). **b,f**: Examples for the seasonal evolution of snow stratigraphy over one winter season in the reference period (b) and in the end of century period (f). **c,g**: Illustration of oberserved layers consisting of precipitation particles, persistent grain types, and rounded grains (c), and refrozen melt form layers (g). (Picture credits: Stephan Harvey (c), Alec van Herwijnen (g))

To investigate the impact of snow stratigraphy, we focused our analysis on four classes of snow grain types grouped according to their microstructural properties (Fig. 5) - precipitation particles, rounded grains, persistent grain types, and melt forms (Table 1). Weak layers associated with dry-snow avalanches (Fig. 5c) typically consist of low-density precipitation particles that cause instabilities during a storm (Bair et al., 2012), or weakly-bonded persistent grain types that form under specific meteorological and snowpack conditions following snow deposition (Birkeland, 1998) and can cause instabilities for prolonged periods. Rounded grains form in dry snowpacks with limited temperature differences, and are generally well bonded. Melt forms indicate wet-snow conditions resulting from liquid water infiltration into the snowpack, i.e. from surface melt water or rain. Although water infiltration can destabilize the snowpack and result in wet-snow avalanches (Conway and Raymond,

1993; Savage et al., 2018), the subsequent formation of crusts after refreezing typically stabilizes the snowpack in the long term (Fig. 5g).

The following results are presented only for the strongest emission scenario (RCP8.5), but changes under other scenarios show similar trends, albeit less pronounced. Our simulations indicate increasing prevalence of melt forms in the snowpack towards the EOC, while persistent grain types will become less common. At the WFJ2 station, melt forms will replace persistent grain types as the most common grain type in DJF snowpacks around mid-century (red and blue lines in Fig. 5a), a trend also observed at all other stations by the EOC (Fig. 5d and 5e). Therefore, the likelihood of forming persistent weak layers associated with dry-snow avalanches will decrease. This decrease is rather intuitive, as persistent grain types primarily form in cold conditions (Colbeck, 1982), and is in line with results of a recent sensitivity study on snow instability (Richter et al., 2020). In contrast, the relative proportion of rounded grains and precipitation particles will remain roughly the same. Since snow depth will generally decrease (Appendix C, Fig. C2e-g), this implies a reduction in the total thickness of precipitation particle layers, a direct consequence of the decrease in the amount of precipitation falling as snow (Appendix C, Fig. C2i-k).

Overall, changes in snow stratigraphy suggest that the occurrence of dry-snow avalanches will decline, as new snow is less likely to fall on a snowpack with persistent weak layers. In contrast, wet-snow avalanches will become more frequent, as the snowpack is more frequently wetted and weakened by melt-water or rain (projections show an up to three-fold increase in 10 mm rain-on-snow events by the EOC under RCP8.5). This is illustrated by the snowpack examples shown in Fig. 5b,f, where the snowpack from the REF period mostly consists of persistent and rounded grains, while the EOC snowpack mostly consists of melt forms. Changes in snow stratigraphy thus offer a coherent explanation for the projected changes in dry- and wet-snow avalanche activity that cannot solely be attributed to a reduction in solid precipitation and snow depth.

## 5  Discussion

Our results show that above the current treeline in the Swiss Alps (typically 2100 to 2300 m a.s.l.;  Gehrig-Fasel et al., 2007), climate change will lead to drastic changes in avalanche types, yet not necessarily to a substantial decrease in the overall number of AvDs (Fig. 3). At these elevations, a decrease in overall avalanche activity by 20-40% by the end of the 21$^{st}$ century is only projected under the strongest emission scenario RCP8.5, while for the other scenarios the decrease is mostly below 10%.

Unlike previous studies relying on bulk snow cover properties or simple snow instability metrics (Lazar and Williams, 2008; Castebrunet et al., 2014; Katsuyama et al., 2022), we used machine learning models to classify dry- and wet-snow AvDs from simulated snow stratigraphy and meteorological variables. This enabled us to comprehensively evaluate the impact of different climate scenarios on snow avalanche activity, investigate changes in seasonality, and highlight causes of projected changes beyond a mere decrease in snow depth or shortening of the snow season. Projections show that dry- and wet-snow avalanche activity respond differently to climate change, and that the overall decrease in dry-snow avalanche activity during DJFMAM is partially compensated by an increase in wet-snow avalanche activity. The strong projected decline in overall avalanche activity at lower-elevation stations (<2200 m a.s.l.) is in line with other studies (Castebrunet et al., 2014; Giacona et al., 2021) and our

projections confirm that under unmitigated emissions, snow avalanches will eventually disappear from progressively higher elevations. While other studies suggested a warming-induced increase in avalanche activity at high elevations (Lavigne et al., 2015; Ballesteros-Cánovas et al., 2018), our results do not confirm this trend, as despite the increase in wet-snow AvDs, the overall avalanche activity during DJFMAM either remains constant or decreases. This discrepancy between our findings for high-elevation stations and those of Lavigne et al. (2015) for high-elevation regions in the French Alps may be attributed to 1) the inadequate representation of extreme events in our model chain, 2) differences in the time periods examined (past observations versus future projections), or 3) data inhomogeneity in the avalanche records used by Lavigne et al. (2015), which could result in misleading conclusions. The study by Ballesteros-Cánovas et al. (2018), based on tree ring analysis from a single slope in the Western Himalayas, is not expected to be representative of conditions in the European Alps. Thus, further research is needed to better understand the impact of climate change on avalanche activity at high elevations.

Regarding the projected changes in snow stratigraphy, the warming-induced earlier appearance of melt forms and crusts and the resulting earlier onset of wet-snow avalanche activity are rather intuitive. Previous studies have found similar trends in projections of wet-snow avalanche activity (Lazar and Williams, 2008; Castebrunet et al., 2014) and snowpack simulations based on observations (Reuter et al., 2022). The projected decline in the proportion of persistent grain types within the DJF snowpack is however less straightforward to anticipate. Although the projected decrease in snow depth might suggest favorable conditions for faceting, the predominant influence appears to be the warmer air temperatures. Enhanced temperatures hinder the formation of weak layers by directly affecting the temperature gradient across the snowpack, primarily through the warming of the snow surface via sensible heat transfer from the warmer air. This increase in surface temperature reduces the temperature gradient between the snow surface and the base of the snowpack. A weaker temperature gradient results in less faceting of snow crystals (e.g.; Colbeck, 1987). In a sensitivity study, Richter et al. (2020) demonstrated that higher air temperatures resulted in a lower proportion of layers with faceted crystals. Interestingly, Katsuyama et al. (2022) also found a decline in the proportion of persistent weak layers based on climate projections for Northern Japan. To further improve our understanding of future avalanche activity, an important next step would be to analyze changes in the predominant grain types of the simulated weakest layers identified by the dry-snow instability model (Mayer et al., 2022), distinguishing between persistent and non-persistent weak layers. Moreover, analyzing the depths of the identified weak layers could yield information on changes in potential avalanche size.

Our projections of future avalanche activity are based on flat field simulations of snow stratigraphy. Typically, during peak winter (December to February), snow profiles from flat study plots are fairly representative of the snow stratigraphy on west-, north- and east-facing slopes. During this period, these slopes and flat sites receive relatively little incoming shortwave radiation, resulting in similar faceting processes. Jamieson et al. (2007) have demonstrated that trends of stability indices from flat field study plots are useful for forecasting dry-snow slab avalanches in surrounding terrain. As the season progresses, differences in radiation become increasingly important. Towards spring, south-facing slopes experience earlier snow melt and thus earlier transition to stable dry-snow and unstable wet-snow conditions compared to north-facing slopes due to differences in incoming radiation. The stratigraphy of flat fields can then be assumed to lie somewhere in between these two extremes. Therefore, we believe that our analysis of avalanche activity based on flat field stratigraphy captures the general trend in future

avalanche activity in the Swiss Alps. Future research should refine our projections by incorporating simulations on virtual slopes with different aspects. In this context, it is important to note that modeled dry-snow instability is most sensitive to precipitation (Richter et al., 2020), which does not vary between aspects except for the critical effects of wind-driven snow redistribution. Accounting for snow drift when analyzing future changes is particularly challenging, as climate projections for wind are typically very uncertain.

There are uncertainties inherent to all parts of our modeling workflow. Air temperature and precipitation are crucial parameters affecting dry- and wet-snow avalanches (Richter et al., 2020; Bellaire et al., 2017). Precipitation likely has the largest uncertainty in our model chain, stemming from the complex downscaling procedure (see Sect. 2.3). Beyond the downscaling methods, uncertainties also arise from the climate models, the avalanche activity models, and the internal variability of the climate system. The substantial variation in projected avalanche activity demonstrates the importance of using various GCM-RCM-chains and RCP scenarios. Nevertheless, all ensemble members consistently projected a decrease in dry-snow AvDs for all stations and an increase in wet-snow AvDs for stations above 2300 m a.s.l. by the EOC, suggesting that our results are robust.

The output of our model chain describes the probability of avalanche release in potential avalanche starting zones at the elevation of the weather stations. Processes along the avalanche path below the starting zones are not accounted for. Projected changes in avalanche activity therefore do not provide information about avalanche size or runout distance. Nonetheless, the projected decrease in snow depths and increased frequency of wetting events (as indicated by the higher proportion of melt forms) suggest that less snow will be present along the avalanche path, and after avalanche release a cold-to-warm flow regime transition will occur more frequently (Köhler et al., 2018). As a result, fewer avalanches are likely to reach valley bottoms since wet snow or bare ground slows down avalanches along their path, which is consistent with an observed increase of avalanche runout elevations during recent decades (Eckert et al., 2013). To quantify future trends in avalanche size and flow regimes, avalanche dynamics models (e.g. Christen et al., 2010; Li et al., 2021) should be driven with climate projections as demonstrated by Ortner et al. (2023).

We emphasize that the presented results focus on changes in mean avalanche activity. The applied downscaling method does not fully account for changes in extreme precipitation, and the avalanche activity models do not predict the magnitude or intensity of the activity. Furthermore, by averaging results over 30-year periods (Fig. 3 and 4), inter-annual variations are averaged out. However, considering the results on a per-winter basis, there is substantial inter-annual variability in the number of AvDs (Appendix D, Fig. D3). This is not surprising, as avalanche activity is determined by the sequence of single weather events over the snow season rather than long-term climate means. In all climate scenarios, some winters at the EOC are projected to have minimal or no dry-snow AvDs, a phenomenon not observed during the reference period. However, with rising wet-snow avalanche activity, the overall likelihood that winters will exceed the median number of AvDs during the reference period will range from roughly 40% for RCP2.6 to 20% for RCP8.5 (Fig. 6). Thus, despite significant warming, some winters will still experience high avalanche activity. To accurately project changes in extreme avalanche cycles, it will be necessary to use dynamically downscaled climate projections that resolve extreme precipitation events in mountain regions (e.g. Schär et al., 2020; Lucas-Picher et al., 2021).

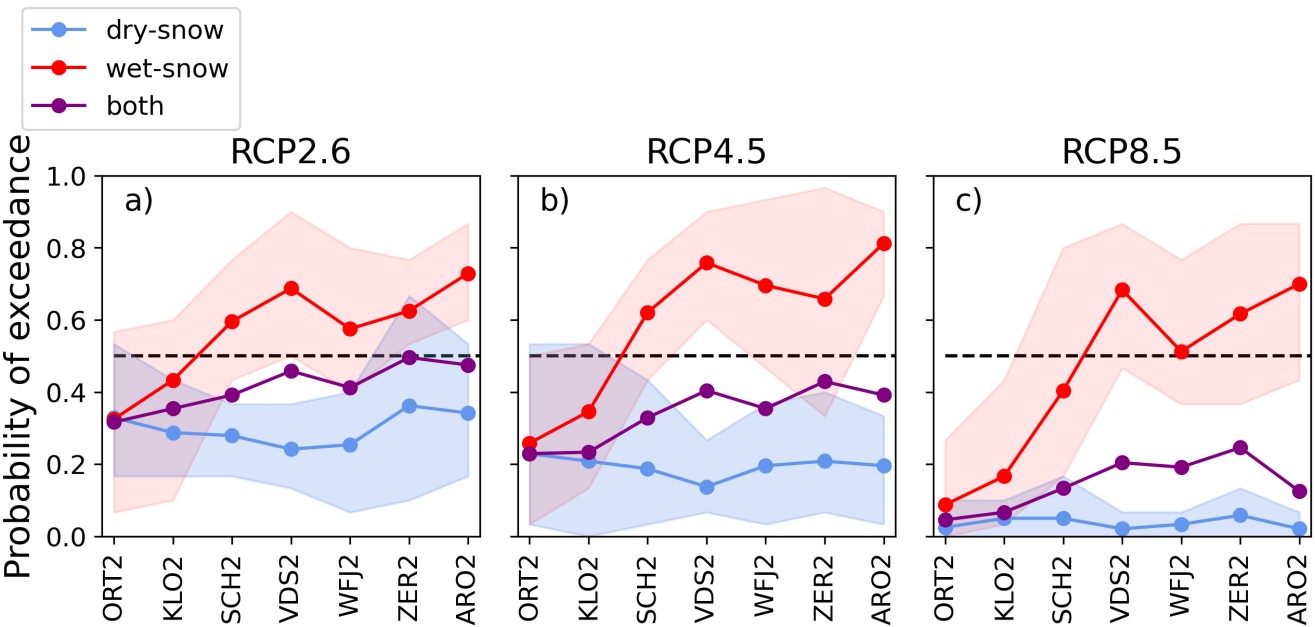

**Figure 6.** Fraction of seasons in the EOC period (2070-2099) that exceed the median number of AvDs per season (DJFMAM) in the REF period regarding dry-snow (blue), wet-snow (red) and dry- or wet-snow avalanche activity (purple) for the seven stations and different emission scenarios RCP2.6 (a), RCP4.5 (b) and RCP8.5 (c). Markers show the multi-model mean, shaded areas correspond to the full model spread. Markers between individual stations are connected for illustration purposes. Stations are arranged in order of increasing elevation, from the lowest (ORT2, 1824 m a.s.l.) to the highest (ARO2, 2847 m a.s.l.).

Projected changes in avalanche activity call for reviewing current avalanche risk mitigation procedures (Bründl and Mar-
greth, 2021). Avalanche risk is determined by the exposure and vulnerability of an object to a specific avalanche hazard, where
the latter is a function of the likelihood of triggering and the destructive size of the avalanche (Bründl et al., 2009). In our
analysis, we focused on the influence of climate change on the likelihood of triggering, and as such we investigated only one
component of avalanche risk. Decreasing dry-snow avalanche frequencies as projected by our models combined with a possible
reduction in avalanche runout demonstrated in other studies (Eckert et al., 2013; Ortner et al., 2023) imply that hazard mapping
procedures may need to be revisited. Nevertheless, updating existing hazard maps will demand more refined projections that
accurately depict extreme avalanche events as outlined above.

Avalanche forecasting may become more challenging, as patterns of avalanche activity will change and unexpected, out-
of-season events may become more frequent. Experience gained so far may prove insufficient for forecasting and managing
some future events, such as wet-snow avalanches during the high-winter season (DJF). Addressing such unfamiliar situations
will require increased awareness and preparedness among avalanche forecasters and safety personnel. In this context, reli-
able avalanche prediction models, such as those used in this study or other recent alternatives (e.g. Pérez-Guillén et al., 2022;
Viallon-Galinier et al., 2023), will become increasingly important. Finally, the anticipated decrease in the likelihood of persis-

tent weak layer formation implies a reduced hazard potential with respect to accidental avalanches triggered by recreationists. Previous studies have indicated that most avalanches triggered by recreationists fail on persistent weak layers (Schweizer and Jamieson, 2001; Techel et al., 2015). However, the frequency of human-triggered avalanches depends on the number of recreationists in the backcountry and on their behaviour. These factors may obscure the climate-related decrease in hazard potential for dry-snow avalanches resulting in stationary accident rates (Strapazzon et al., 2021).

## 6  Conclusions

We quantified changes in avalanche activity throughout the 21st century for seven sites located at mid-to-high elevations (1800-2900 m a.s.l.) in the Swiss Alps using a comprehensive modeling approach. This method integrated downscaled climate scenarios, physics-based simulations of snow stratigraphy, and machine learning models to classify days with or without avalanche occurrence. Our findings revealed elevation-dependent patterns of change, indicating a decrease in the number of dry-snow AvDs alongside an increase in wet-snow AvDs at elevations above 2300 m a.s.l. for all emission scenarios. For the highest emission scenario RCP8.5, the number of dry-snow AvDs during DJFMAM is projected to continuously decline throughout the 21st century for all stations, reaching a relative decrease of 45-65% by the EOC. This change is attributable to the decrease in snowfall events, but also to the decrease in layers with persistent grain types. On the other hand, projections under the same emission scenario RCP8.5 indicate that the number of wet-snow avalanche days during DJFMAM at elevations above 2300 m a.s.l. will rise and peak during the latter half of the century, showing a relative increase of 10-30%. Simulations indicated a shift of wet-snow activity to earlier winter months, driven by the increased frequency of wetting events. These results challenge simplistic assumptions about a consistent decline in avalanche occurrences due to diminishing snow cover, highlighting the importance to distinguish between dry- and wet-snow avalanches.

While our results focus on the Swiss Alps, we anticipate changes in average avalanche activity will be comparable across much of the European Alps, and other regions with similar avalanche climates and elevations relative to the average snow line. For example, this includes regions like the Columbia Mountains in western Canada, which are characterized by a transitional snow climate (Shandro and Haegeli, 2018). However, further simulations will have to confirm the transferability of our results. Additionally, future research should focus on changes at higher elevations, where measurements are currently lacking, account for changes in extreme precipitation, and investigate the impact on avalanche size. Dynamical downscaling methods will therefore be required to more accurately project changes in extreme weather events in mountain regions, and avalanche dynamics models should be employed to assess changes in avalanche flow regimes and impact pressures. While such detailed models will entail greater computational cost, they will refine our results and provide essential understanding for developing robust avalanche forecasting and mitigation procedures in a warming climate.

As avalanche activity is projected to change, avalanche risk mitigation procedures should be revisited regularly. At present, procedures such as hazard mapping assume stationary conditions when assessing the probability of extreme events. In view of the changing climate, i.e. transient rather than stationary conditions, it seems reasonable to review the procedures more frequently than in the past, for example every ten years. Furthermore, given the projected seasonal changes in avalanche

activity, so-called unexpected events may occur more frequently, calling for increased awareness and agile risk mitigation procedures.

*Data availability.* We will establish a repository where projections of avalanche activity will be accessible upon publication.

# Appendix A: Climate scenarios

**Table A1.** Combinations of GCM, RCM, and horizontal resolution of the eight climate model chains from the EURO-CORDEX ensemble used in this study.

| GCM | RCM | EUR-11 (0.11°, ~12 km) | EUR-44 (0.44°, ~50 km) |
|---|---|---|---|
| ICHEC-EC-EARTH | DMI-HIRHAM5 | x | |
| MOHC-HadGEM2-ES | SMHI-RCA4 | | x |
| ICHEC-EC-EARTH | SMHI-RCA4 | x | x |
| MIROC-MIROC5 | SMHI-RCA4 | | x |
| MPI-M-MPI-ESM-LR | SMHI-RCA4 | | x |
| NCC-NorESM1-M | SMHI-RCA4 | | x |
| MOHC-HadGEM2-ES | KNMI-RACMO22E | | x |

**Table A2.** Settings used in the MELODIST library, see Förster et al. (2016) for details.

| Variable | Setting |
| --- | --- |
| Air temperature | mean_course_mean |
| Relative humidity | equal |
| Incoming shortwave radiation | pot_rad |
| Precipitation | cascade |
| wind speed | random |

## Appendix B: Historical time series of meteorological data

To downscale the CH2018 climate scenarios to the specific locations of the seven IMIS AWS, robust historical time series of meteorological data corresponding to these stations were necessary. These time series should encompass all the essential parameters for driving the snow cover model SNOWPACK (precipitation, air temperature, incoming shortwave radiation, wind speed, relative humidity), aligning with the variables available in the climate scenarios. For the station WFJ2, a quality-checked data set of meteorological measurements spanning the years 1999-2017 was available (Wever, 2017). For the other stations, we implemented the following steps using the data measured at the IMIS AWS over the time period 2000-2022:

1. Pre-processing the raw data: First, each IMIS AWS was associated with a nearby SwissMetNet AWS. Data gaps in summer due to IMIS station maintenance were then manually filled with data from the SwissMetNet AWS. Snow depth measurements were cleaned using a high-frequency filter and a second filter implemented in the MeteoIO library (Bavay and Egger, 2014). Remaining gaps or peaks in snow depth were manually cleaned through linear interpolation. Inconsistent values of snow surface temperature, i.e. positive values when the snow depth was larger than 5 cm, were manually replaced with a value of -1 °C. A full list of all manual corrections is given in (Michel et al., 2023).

2. Deriving solid precipitation: Precipitation at the IMIS AWS is only measured with unreliable rain gauges that are not heated. We therefore estimated solid precipitation values from SNOWPACK simulations driven with measured snow depth (Wever et al., 2015), employing an empirical relationship for new snow density as a function of air temperature, relative humidity and wind speed (Schmucki et al., 2014).

3. Merging precipitation data: Solid precipitation estimated with SNOWPACK was merged with precipitation data from the nearby SwissMetNet AWS using an air temperature threshold of 2 °C. Below this threshold, we used the solid precipitation from SNOWPACK, above we used precipitation measurements from SwissMetNet. We opted for this approach as the precipitation data from SwissMetNet is reliable solely for liquid precipitation, given that the heated rain gauges encounter undercatch issues during snowy conditions. SNOWPACK forced with measured snow depth, on the other hand, can only retrieve solid precipitation.

4. Incoming shortwave radiation: At an IMIS AWS, incoming shortwave radiation (ISWR) is not measured. Instead, reflected shortwave radiation measurements are used to force SNOWPACK and ISWR is derived using an estimated snow albedo. In summer, however, the ground albedo is more variable and mostly unknown, resulting in false estimates of ISWR. The downscaling schemes yet require time series spanning whole years. To circumvent this problem, we used ISWR measurements of the nearby SwissMetNet AWS.

Note that due to uncertainties in the precipitation reconstruction and the simple transfer of ISWR, these time series should be considered as representative for the area, rather than for the exact location of the AWS.

  **Appendix C:  Validation**

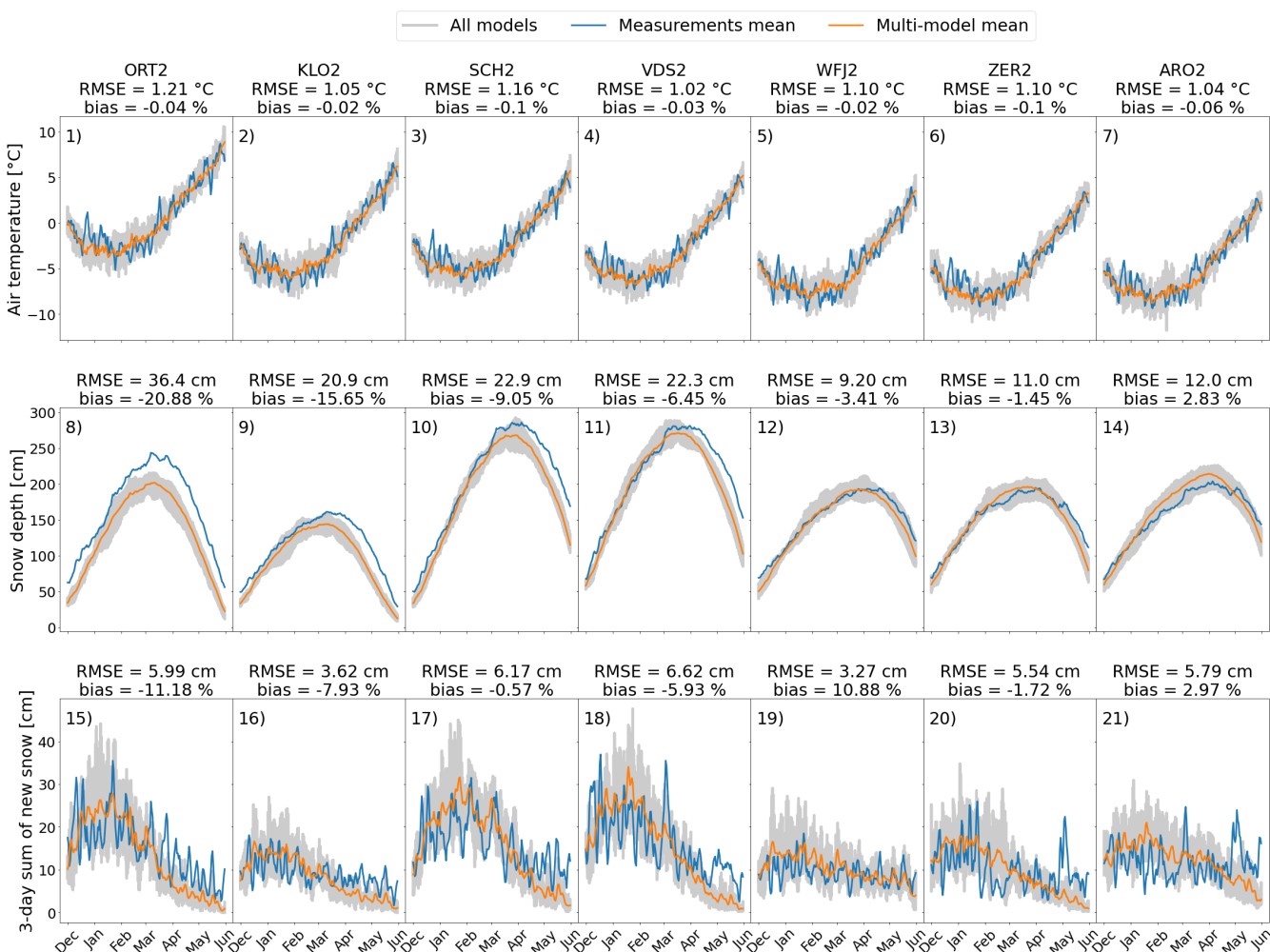

**Figure C1.** Comparison of measured and modeled meteorological variables for each day for the months December to May over the period 2001-2022 for the seven AWS used in this study (1 column per AWS). Blue lines: daily mean values of the AWS measurements and resulting parameters from SNOWPACK simulations. Orange lines: daily mean of the simulated variables from downscaled climate models and resulting parameters from SNOWPACK simulations for the RCP8.5 scenario. The gray-shaded areas indicate the spread of the eight climate ensemble members. Air temperature (1-7), snow depth (8-14), three-day sum of new snow height (15-21).

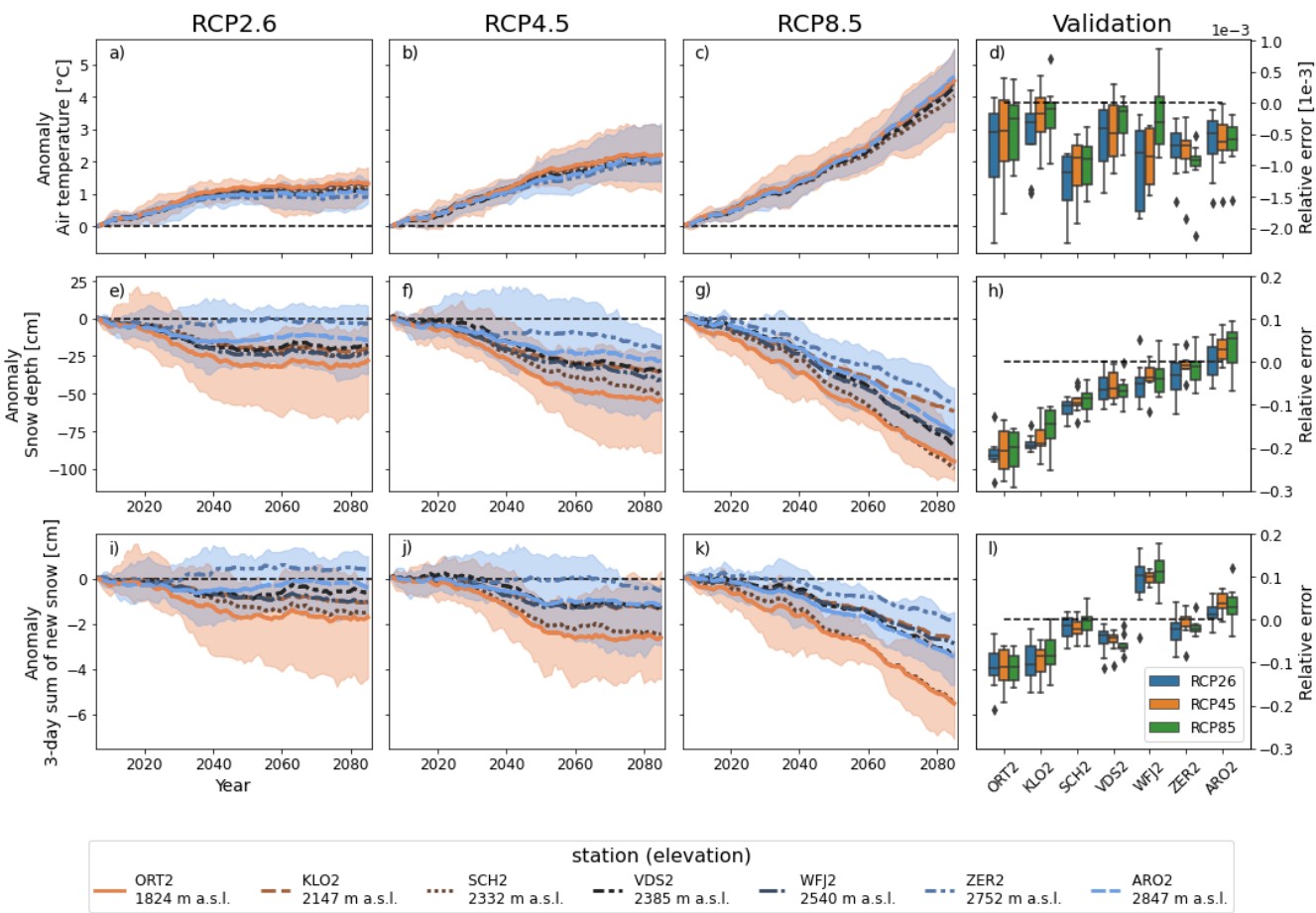

**Figure C2.** Anomalies of air temperature (a,b,c), snow depth (e,f,g), three-day sum of new snow height (i,j,k) compared to the REF period for emission scenarios RCP2.6, RCP4.5 and RCP8.5. Lines show the multi-model mean of the respective variable (centered 30-year moving mean) for seven AWS across the Swiss Alps. Shaded areas indicate the spread of the eight ensemble members for the highest and lowest AWS. Station identification is facilitated by colors and line styles specified in the bottom legend. Validation of downscaled climate models (d,h,l) comparing variables based on climate simulations and resulting SNOWPACK output with AWS measurements and resulting SNOWPACK output for the winter seasons 2000/2001-2021/2022. The box and whiskers show the variability within the eight ensemble members, where the line indicates the median, the box extends from the first to the third quartile, the whiskers extend to 1.5 the interquartile range beyond the box limit, and the black diamonds show outliers.

## Appendix D: Projections of avalanche activity

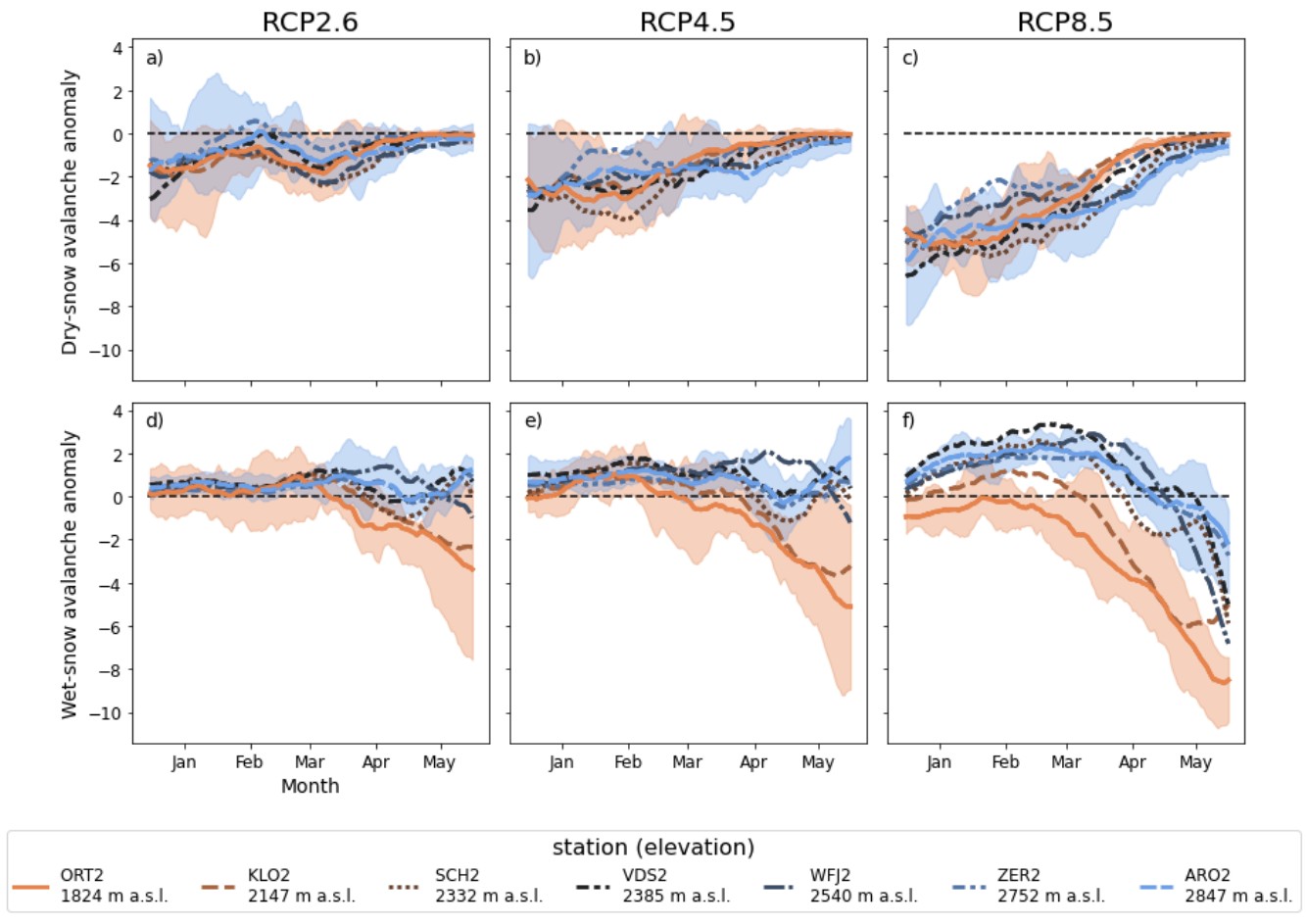

**Figure D1.** Mean monthly dry- and wet-snow avalanche anomaly for the 2070-2099 period compared to the 1991-2020 period for RCP2.6 (a,d), RCP4.5 (b,e) and RCP8.5 (c,f). Lines show the multi-model mean for seven AWS across the Swiss Alps. Shaded areas indicate the spread of the eight ensemble members for the highest and lowest AWS. Station identification is facilitated by colors and line styles specified in the bottom legend.

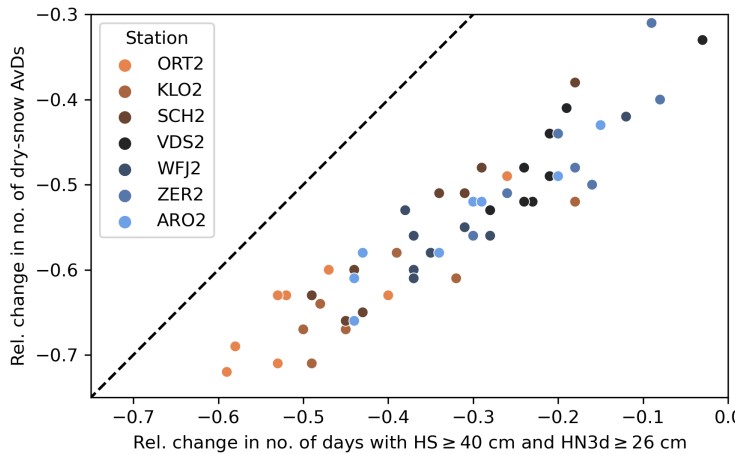

**Figure D2.** Relative changes ((REF-EOC)/REF) in the number of dry-snow AvDs under emission scenario RCP8.5 versus relative changes in the number of days with snow depth (HS) of at least 40 cm and three-day sum of new snow height (HN3d) of at least 26 cm under RCP8.5. The threshold HN3d = 26 cm must be exceeded for a day to be classified as AvD by the HN3d-model in Mayer et al. (2023) which represents one component of the dry-snow model used in this study. Dots refer to individual climate model chains (8 per station) and colors represent the different stations. The black dashed line represents the identity line.

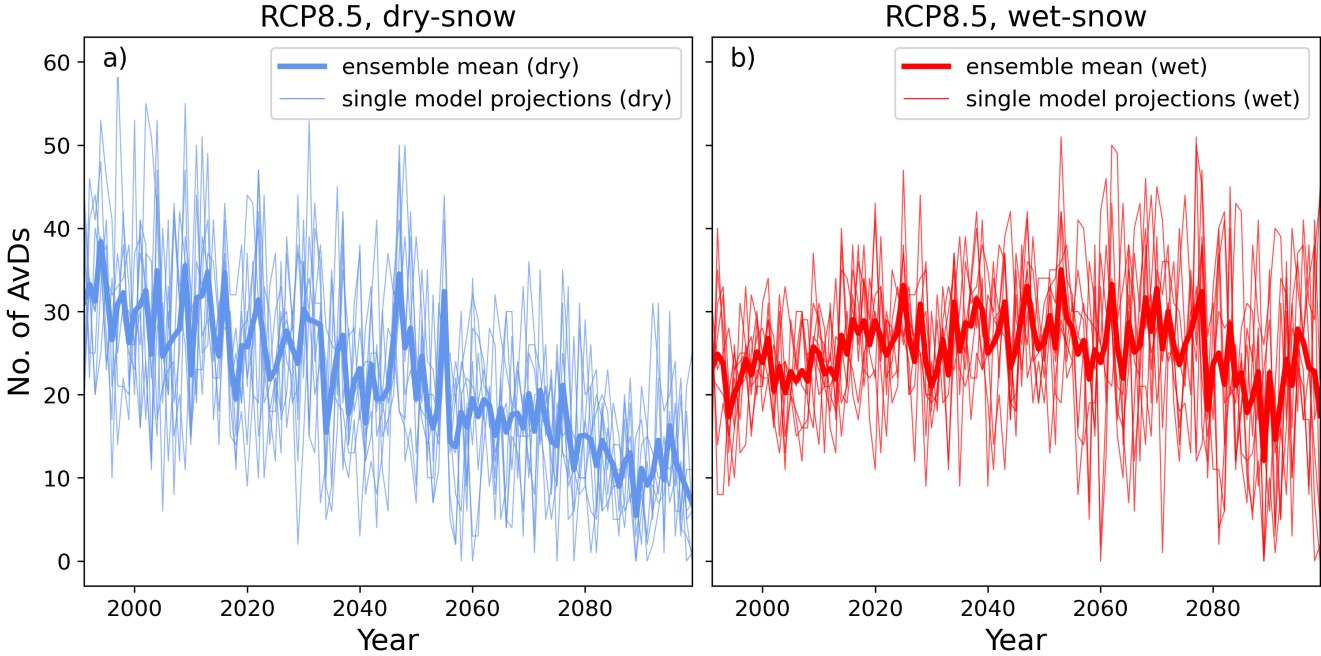

**Figure D3.** Projected number of (a) dry-snow and (b) wet-snow AvDs for the station WFJ2 (2540 m a.s.l.) on a yearly resolution for the timespan 1991-2099. Thin lines indicate projections by individual model chains and the bold line shows the multi-model mean.

*Author contributions.* AH and JS initiated this study, AM conducted the downscaling of climate projections, SM and BR ran the SNOW-PACK simulations, SM and MH applied the avalanche activity models and processed and analyzed the simulations. SM prepared the manuscript with contributions from all co-authors.

*Competing interests.* At least one of the (co-)authors is a member of the editorial board of The Cryosphere.

*Acknowledgements.* This work was partly funded by the WSL research program CCAMM (Climate Change Impacts on Alpine Mass Movements). We thank Nicolas Eckert for providing useful advice on our manuscript. We thank the editor, Edward Bair, and two anonymous reviewers for their thorough reviews which have contributed to improving our manuscript.

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
