# Peer review of "Impact of climate change on snow avalanche activity in the Swiss Alps"

_EGUsphere, 2024_

## Author Comment (AC1)

**Reply to Reviewer #1**

We thank reviewer #1 for the constructive review of our manuscript, which we greatly appreciate. In the following, we address each of the points raised. Black text indicates the reviewer's comments. Below we describe (in blue) how we will address each comment when revising the manuscript.

**General comments**

This study examines future avalanche activity in the Swiss Alps with a modelling approach that combines downscaled climate projections, a detailed snow stratigraphy model, and machine learning models. The models predict the number of days with dry and wet snow avalanches under different climate scenarios at 7 high-elevation locations, finding an overall decrease in dry snow avalanches and an increase in wet snow avalanches. These trends are interpreted by showing how the snowpack structure will change at different elevations.

The overall quality of the manuscript is high, with a clear presentation of the methods and results and interesting discussion and conclusions. The study design combines several state-of-the-art methods to investigate a significant topic that fits well within the scope of The Cryosphere. I recommend publication after addressing the following comments.

**Specific comments**

- **Climate warming versus climate change:** From what I understand "climate warming" primarily refers to the temperature component of "climate change", as opposed to the entire climate system (precipitation, feedback loops, etc.). Were the authors deliberate in how they used each term throughout the manuscript (e.g., "climate warming" in the title suggests the study focuses on the response to warming temperatures)?

  Thank you for pointing out the differences between these notions. We used the term "climate warming" as we consider rising temperatures to be the primary driver of all other changes. However, based on your comment, in the revised version of the manuscript, we will consistently use the term "climate change".

- **Explanation of climate models and data:** The climate data used for the study should be explained in more detail. The SNOWPACK model is thoroughly explained, but there is comparatively little information about the EURO-CORDEX model and CH2018 datasets, especially for readers unfamiliar with these products. Please briefly introduce and define GCMs and RCMs and explain the differences between the three RCP scenarios. Listing the specific GCM/RCM models in Table B1 has minimal meaning without explaining what they are and how they differ. Also, specifically for this study, it would be valuable to comment on how well the CH2018 datasets resolve fluctuating weather systems over a season and whether they are appropriate for predicting realistic snowpack stratigraphies. Do they produce smoothed average values, or do they capture realistic storms interspersed with high-pressure weather? For example, warming-related intensification of heavy snowfall is mentioned in line 38, would this be reflected in the

data? The importance of resolving extreme weather events is addressed in the discussion section, but commenting on these aspects earlier would help readers interpret the results.

We will provide more detailed information on the climate scenarios and underlying models in the revised version of the manuscript. In particular, we will point out that the CH2018 data set does provide transient daily time series of meteorological variables rather than only average values. The CH2018 climate scenarios were statistically downscaled from the coarser RCM simulations of the EURO-CORDEX ensemble to automatic weather stations (AWS) using quantile mapping (QM). This means that, for example, if the daily precipitation sum on a given day in the corresponding RCM grid cell equals the 70th percentile of the reference period climate simulations, then the daily precipitation sum at a particular station is set to the 70th percentile of the daily precipitation observed at that station during the reference period. In principle, this implies that the downscaling preserves realistic weather patterns and the downscaled data capture realistic storm events that are present in the RCM simulations if those events are represented in the reference period. However, QM assumes that the relation between conditions in the grid cell and the station is stationary, which might not hold true under changing climatic conditions. Moreover, the QM transfer function is subject to uncertainties due to the limited number of years considered for the calibration, in particular for high and low quantiles. Furthermore, the extremes that have not been observed in the reference period are mapped using a constant extrapolation of the biases for the 1st and 99th percentiles. As you suggested, we will comment on these aspects in the methods section of the revised manuscript.

- **Quantile mapping method:** Please explain the basic principles behind quantile mapping to provide an overview of its function and application. Was the goal bias correction, downscaling, elevation adjustment, etc., and why was it chosen over alternative statistical methods?

  Quantile mapping (QM) is an established statistical method used to transform the distribution of modelled data so that its statistical properties align more closely with observed data. When applied in a climate modelling context, QM involves building a transfer function that relates quantiles from the modelled distribution of a meteorological variable to the corresponding quantiles in the observed distribution for a given reference period. The transfer function is then applied to bias-correct a transient time series of the simulated variable. QM is thus primarily a bias-correction method but can implicitly serve as a downscaling method if the observed reference data refer to a higher spatial resolution than the climate model (Feigenwinter et al., 2018).

  We used QM to transfer the CH2018 climate scenarios for the SwissMetNet stations to the higher-elevation IMIS stations. Herein, the multivariate quantile mapping approach from Cannon (2018) enabled us to do a bias-correction while restoring intervariable relationships, a clear advantage over alternative statistical methods for bias correction. The CH2018 scenarios for the SwissMetNet stations were themselves obtained by applying the QM approach to the EuroCordex RCM simulations, thus simultaneously bias-correcting and downscaling this data set (see previous answer). We will provide missing information in the revised version of the manuscript.

  We preferred QM over other methods like the delta change method (Michel, 2021) or weather generators (Peleg, 2019) because it can generate transient time series while remaining computationally efficient.

- **Figures:** The overall figures quality is high, but some figures are confusing because they mix multiple data types and axes into a single graphic. It would be worth examining whether any subfigures should be split or omitted. Examples where data/axes do not fit with the rest of the figure include the left column in Fig. 3, the right column in Fig. 4, the mix of subfigures in Fig. 5, and the validation column in Fig. C2. Also, several axes label relative differences as fractions, but the manuscript text discusses them in terms of percentages. Perhaps it would be easier to interpret if the axes were labelled with percentage values (i.e., -10% instead of -0.1)?

  We agree that some figures include a lot of information. However, with the comprehensive figure captions we provide, we think there is no need to change the figures. It does not seem unusual to us to have several panels with different axes in one illustration.

- **Impact of validation findings:** 3b suggests the model chain has a systematic bias towards underpredicting wet avalanche days at low elevations and overpredicting at high elevations. Are there any known reasons for this and could it be corrected? Furthermore, how does this impact the results? Fig. 3 and 4 show distinct trends in wet avalanche activity by station elevation, but are these results exaggerated due to this bias?

  Thank you for this interesting remark on Fig. 2b (not 3b we suppose). As the underlying model chain (downscaling procedure + SNOWPACK + random forest "blackbox" models) is quite complex, it is hard to tell why wet-snow avalanche days are underpredicted at lower elevations. Interestingly, the validation of the simulated snow height shows a similar systematic bias towards underpredicting snow height, with higher relative differences for lower stations (Appendix Fig. C2 h). Lower snow depths also imply fewer days with snow cover and thus fewer potential wet-snow avalanche days. Yet, this relation is not sufficient to explain observed biases, as the validation does not show the same trend for the dry-snow avalanche days (Fig. 2a). Appendix Figs. C1 8-11 reveal that snow depth at the four lowest stations is particularly underestimated after the peak of snow depth in March / April, which is a relevant time period regarding wet-snow avalanche activity. This underestimation of snow depth at the end of the season can either originate from an underestimation of snowfall (Appendix Figs. C1 15-18), or from an overestimation of melt rates due to differences in the input variables and SNOWPACK settings between the simulations based on the climate scenarios and the simulations based on measurements. For example, incoming shortwave radiation in the climate scenarios is obtained from quantile mapping based on shortwave radiation measurements from a nearby SwissMetNet station, as it is not measured at the IMIS stations. This uncertainty in radiation might also contribute to errors in the simulation of the ablation period and thus influence the parameters used as input for the wet-snow random forest model (e.g. the liquid water content). Yet, finding the ultimate reason for the observed trend in the deviations between simulations driven with climate data versus measurements would require more in-depth station-specific analyses considering all uncertainties inherent in the model chain, and goes beyond the scope of our paper.

  In this context, we have noticed that the manuscript is missing a paragraph on the SNOWPACK settings used in the simulations based on measurements for validation. We apologize for this oversight and will add this paragraph in the revised version of the manuscript.

- **Spatial/frequency distribution:** The discussion acknowledges that the model only addressed one component of avalanche risk: the likelihood of triggering. While the importance of destructive size is discussed, the spatial/frequency distribution of avalanches is also a commonly recognized

component. The European Avalanche Warning Services defines avalanche danger based on snowpack stability, frequency distribution, and avalanche size. Similarly, the conceptual model of avalanche hazard defines the "likelihood of avalanches" as a function of "sensitivity to triggers" and "spatial distribution." The spatial component of hazard should be acknowledged and discussed. For example, the results show some elevation trends that may relate to frequency distribution, but in general the machine learning models primarily predicted likelihood of triggering. Could the methods be adapted to further examine this, either with spatially distributed snowpack models (e.g., by aspect), or perhaps training machine learning models to predict distribution?

We agree that the frequency distribution of snow instability is an important element for defining the avalanche danger level in avalanche forecasts. Avalanche forecasts are issued for a certain region. Therefore, it is important to know how many avalanches are to be expected, and of which size. In our study, we simply assess the triggering potential given a certain snow stratigraphy. In other words, there is no spatial component. It is beyond the scope of this study to estimate how the spatial distribution would be affected.

We will address the spatial component of hazard in the discussion of the revised manuscript. In our view, one crucial part in obtaining reasonable projections of future avalanche activity is the preparation of meteorological input data that represent the conditions at the resolution of interest. To obtain spatially distributed projections of avalanche activity, high-resolution gridded input data to drive the model chain is necessary. As such projections containing all variables required to drive SNOWPACK were not available, we focused on obtaining climate scenarios for seven stations at different elevations to obtain a first estimate on the trend of future avalanche activity in the Swiss Alps. In a next step, these projections could be refined by analyzing results for different aspects (see also answer to the editor).

- **References:** The manuscript is well cited, but perhaps the reference list is a little long at 5 full pages. Also, I think Verfaillie et al. (2018) is very relevant to this study, and perhaps the recently published review by Eckert et al. (2024).

  Thank you for mentioning these studies which we will cite in the revised version. As in our view there are no useless references in our reference list, we would like to keep all existing references.

**Technical comments**

- Line 110: Appendix B is mentioned before Appendix A. Consider reordering.

  Thank you for noting this. We will reorder the appendices.

- Line 178: How was three-day sum of new snow height calculated and was it done the same way for SNOWPACK output and AWS measurements?

  The three-day sum of new snow height provided by SNOWPACK is calculated as the sum of the three consecutive values of the height of new snow within 24 hours. The AWS measurements provide snow depth only and do not measure new snow height. Hence, for this case, SNOWACK

was driven with snow depth measurements, and new snow height was calculated taking into account settlement of the snowpack. For the simulations driven with the downscaled climate model output, SNOWPACK was forced with precipitation and the height of new snow was again computed considering settlement. We will add this information in the revised version.

- Line 181-183: It seems odd to introduce a figure in an appendix before a figure in the main body. I would consider reordering (assuming Fig. 2 is more relevant than Fig. C1).

  We will reorder the two sentences.

- Line 198: "and with AWS measurements" would better explain relative differences than "or with AWS measurements".

  As the sentence refers to two different types of forcings ("SNOWPACK simulations either forced with the downscaled climate models or with AWS measurements"), we believe that "or" is the better choice in this case. To enhance clarity, we will rewrite the sentence to "To validate the complete model chain, including both dry- and wet-snow avalanche activity models, we calculated the seasonal number of avalanche days (AvDs) using SNOWPACK simulations. These simulations were forced with either downscaled climate models or AWS measurements, and we then computed the relative difference between the resulting seasonal numbers of AvDs."

- 3: Do you have any idea why the ensemble of climate models resulted in more spread for wet avalanche days than the dry avalanche days? Was the wet avalanche machine learning model more sensitive to a specific input?

  Thank you for the interesting remark. To answer this question with confidence, a detailed sensitivity analysis comparing both avalanche activity models would be necessary. As this would go beyond the scope of our study, we can only speculate here. The most important meteorological parameter for the dry-snow model is precipitation. For the wet-snow model, on the other hand, radiation and temperature are explicitly included as input features, and precipitation still plays a major role as it drives the length of the snow season and thus the potential length of the wet-snow avalanche season. The dependence of the wet-snow model on the combination of different meteorological parameters can thus be assumed to be stronger compared to the dry-snow model, leading to larger overall variability.

- Line 265: An aside comment… Could you split the dry snow avalanche days by weak layer type to count the number of days with avalanches on persistent versus non-persistent grain types? What other information about the avalanches could be derived from snowpack models (avalanche problems, aspect-trends, etc.)?

  Identifying the weak layer grain type would indeed be an interesting next step. However, this is not as straightforward as one might think, for example, we would have to decide whether to consider all layers exceeding the instability threshold, or just the deepest one or the one with the highest instability value. We therefore did not include this in our analysis. As mentioned above, analyzing trends for different aspects would be an interesting next step. Moreover, analyzing the depths of the weak layers identified by the dry-snow instability model could yield interesting information on changes in potential avalanche size.

- Line 297: "and despite" instead of "as despite".

  We intended to express causality with the use of "as" in this case.

**References**

Eckert, N., Corona, C., Giacona, F. et al. Climate change impacts on snow avalanche activity and related risks. Nat. Rev. Earth Environ., 5, 369, https://doi.org/10.1038/s43017-024-00540-2, 2024.

Verfaillie, D., Lafaysse, M., Déqué, M., Eckert, N., Lejeune, Y., and Morin, S.: Multi-component ensembles of future meteorological and natural snow conditions for 1500 m altitude in the Chartreuse mountain range, Northern French Alps, The Cryosphere, 12, 1249–1271, https://doi.org/10.5194/tc-12-1249-2018, 2018.

**References**

Cannon, A. J.: Multivariate quantile mapping bias correction: an N-dimensional probability density function transform for climate model simulations of multiple variables, Climate Dynamics, 50, 31–49, https://doi.org/10.1007/s00382-017-3580-6, 2018.

Feigenwinter, I., Kotlarski S., Casanueva A., Fischer A.M., Schwierz C., and Liniger M. A.: Exploring quantile mapping as a tool to produce user-tailored climate scenarios for Switzerland. Technical Report MeteoSwiss No. 270, MeteoSchweiz, 2018.

Michel, A., Sharma, V., Lehning, M., and Huwald, H.: Climate change scenarios at hourly time-step over Switzerland from an enhanced temporal downscaling approach. Int J Climatol., 41: 3503–3522, https://doi.org/10.1002/joc.7032, 2021

Peleg, N., Molnar, P., Burlando, P. and Fatichi, S.: Exploring stochastic climate uncertainty in space and time using a gridded hourly weather generator. Journal of Hydrology, 571, 627–641. https://doi.org/10.1016/j.jhydrol.2019.02.010, 2019

---

## Author Comment (AC3)

**Reply to the Editor**

We thank the Editor for his comments. Please find our replies and how we plan to revise the manuscript in blue below.

I have read your submission and am starting the review process. I have two points that I suggest addressing during the revision.

This study is biased towards Swiss avalanches, which is fine, but should be acknowledged. For example, layering and the prevalence of avalanches that fail on old snow layers varies throughout the world. In Switzerland or Colorado USA, old snow avalanches dominate. In California USA or coastal British Columbia Canada, avalanches mostly involve the new snow.

Our study refers to the Swiss Alps, as stated in the title of our manuscript. We will further elaborate on the representativity of our study and describe that the selected sites cover different climate zones in the Swiss Alps in the Discussion section.

Flat sites are a poor representation of avalanche starting zones. Why weren't virtual slopes, as in Mayer et al. (2023), used? How was the shortwave & longwave radiation balance, which is critical for facet growth, adjusted to steep north facing slopes? Was any comparison performed between the SNOWPACK profiles at the flat sites and in avalanche starting zones? If the snowpack is consistently shaded throughout the winter, one can still expect facet formation under warming, maybe even increases with the thinner snowpack, given the negative radiation balance. Similarly, won't melting occur much earlier and more frequently on the flat study plots than in the starting zones?

We agree that analyzing slope simulations would provide added value for understanding future avalanche activity. Nevertheless, we cannot generally agree that flat sites are a poor representation of avalanche starting zones. For instance, Jamieson et al. (2007) have demonstrated that trends of stability indices from flat field study plots are useful for forecasting dry-snow slab avalanches in surrounding terrain. Moreover, data from SNOWPACK simulations at flat fields proved to be useful input for numerical models of avalanche forecasting (e.g. Pérez-Gullién et al., 2022). Typically, during peak winter (December to February), snow profiles from flat study plots are fairly representative of the snow stratigraphy on west-, north- and east-facing slopes. During this period, these slopes and flat sites receive relatively little incoming shortwave radiation, resulting in similar faceting processes. As the season progresses, differences in radiation become increasingly important. Towards spring, south-facing slopes experience earlier snow melt and thus earlier transition to stable dry-snow and unstable wet-snow conditions compared to north-facing slopes due to differences in incoming radiation. The stratigraphy of flat fields can then be assumed to lie somewhere in between these two extremes. Therefore, we believe that our analysis of avalanche activity based on flat field stratigraphy captures the general trend in future avalanche activity in the Swiss Alps.

In the Discussion section, we will acknowledge that our projections are based on flat field simulations and that future studies could refine these results considering different slope aspects. Herein, it is also

important to note that modeled dry-snow instability is most sensitive to precipitation (Richter et al., 2020), which does not vary between aspects apart from the crucial snow redistribution by wind. Accounting for snow drift when analyzing future changes may be very challenging, as climate projections of wind are typically very uncertain.

References

Jamieson, J. B., Zeidler, A., and Brown, C.: Explanation and limitations of study plot stability indices for forecasting dry snow slab avalanches in surrounding terrain, Cold Reg. Sci. Technol., 50, 23-34, https://doi.org/10.1016/j.coldregions.2007.02.010, 2007.

Pérez-Guillén, C., Techel, F., Hendrick, M., Volpi, M., van Herwijnen, A., Olevski, T., Obozinski, G., Pérez-Cruz, F., and Schweizer, J.: Data-driven automated predictions of the avalanche danger level for dry-snow conditions in Switzerland, Nat. Hazards Earth Syst. Sci., 22, 2031-2056, https://doi.org/10.5194/nhess-22-2031-2022, 2022.

Richter, B., van Herwijnen, A., Rotach, M.W., and Schweizer, J.: Sensitivity of modeled snow stability data to meteorological input uncertainty, Natural Hazards and Earth System Sciences, 20, 2873–2888, https://doi.org/10.5194/nhess-20-2873-2020, 2020.

---

## Author Response (AR1)

**Reply to the Editor**

We thank the editor Edward Bair for the positive and constructive feedback. Please find below our detailed replies (in blue).

This study is biased towards Swiss avalanches, which is fine, but should be acknowledged. For example, layering and the prevalence of avalanches that fail on old snow layers varies throughout the world. In Switzerland or Colorado USA, old snow avalanches dominate. In California USA or coastal British Columbia Canada, avalanches mostly involve the new snow.

Our study refers to the Swiss Alps, as stated in the title of our manuscript. We now describe that the selected sites cover different climate zones in the Swiss Alps (Sect 2.1, lines 83-85) and have revised the sentence regarding the representativeness of our study in the Conclusions section (lines 447-450).

Flat sites are a poor representation of avalanche starting zones. Why weren't virtual slopes, as in Mayer et al. (2023), used? How was the shortwave & longwave radiation balance, which is critical for facet growth, adjusted to steep north facing slopes? Was any comparison performed between the SNOWPACK profiles at the flat sites and in avalanche starting zones? If the snowpack is consistently shaded throughout the winter, one can still expect facet formation under warming, maybe even increases with the thinner snowpack, given the negative radiation balance. Similarly, won't melting occur much earlier and more frequently on the flat study plots than in the starting zones?

In the revised Discussion section, we now acknowledge that our projections are based on flat field simulations and that future studies could refine these results considering different slope aspects (lines 370-383).

**Reply to Reviewer #1**

We thank Reviewer #1 for the positive and constructive feedback. Please find below our detailed replies (in blue).

**Specific comments**

- **Climate warming versus climate change:** From what I understand "climate warming" primarily refers to the temperature component of "climate change", as opposed to the entire climate system (precipitation, feedback loops, etc.). Were the authors deliberate in how they used each term throughout the manuscript (e.g., "climate warming" in the title suggests the study focuses on the response to warming temperatures)?

  Thank you for pointing out the differences between these notions. We used the term "climate warming" as we considered rising temperatures to be the primary driver of all other changes. However, based on your comment, we now consistently use the term "climate change" in the revised version of the manuscript. To avoid using the word "change" twice in the title, we changed the title to "Impact of climate change on snow avalanche activity in the Swiss Alps".

- **Explanation of climate models and data:** The climate data used for the study should be explained in more detail. The SNOWPACK model is thoroughly explained, but there is comparatively little information about the EURO-CORDEX model and CH2018 datasets, especially for readers unfamiliar with these products. Please briefly introduce and define GCMs and RCMs and explain the differences between the three RCP scenarios. Listing the specific GCM/RCM models in Table B1 has minimal meaning without explaining what they are and how they differ. Also, specifically for this study, it would be valuable to comment on how well the CH2018 datasets resolve fluctuating weather systems over a season and whether they are appropriate for predicting realistic snowpack stratigraphies. Do they produce smoothed average values, or do they capture realistic storms interspersed with high-pressure weather? For example, warming-related intensification of heavy snowfall is mentioned in line 38, would this be reflected in the data? The importance of resolving extreme weather events is addressed in the discussion section, but commenting on these aspects earlier would help readers interpret the results.

  We now provide more detailed information on the climate scenarios and underlying models in the revised version of the manuscript (new Sect. 2.2 "Climate projections CH2018").

- **Quantile mapping method:** Please explain the basic principles behind quantile mapping to provide an overview of its function and application. Was the goal bias correction, downscaling, elevation adjustment, etc., and why was it chosen over alternative statistical methods?

  We included a new paragraph explaining the basic principles behind quantile mapping (Sect. 2.2). We also explained that in the context of our study, quantile mapping was used to correct for biases in the climate model output, implicitly downscaling RCM gridded data to a higher spatial resolution (stations).

We preferred QM over other methods like the delta change method (Michel et al., 2021) or weather generators (Peleg et al., 2019) because it can generate transient time series while remaining computationally efficient.

- **Figures:** The overall figures quality is high, but some figures are confusing because they mix multiple data types and axes into a single graphic. It would be worth examining whether any subfigures should be split or omitted. Examples where data/axes do not fit with the rest of the figure include the left column in Fig. 3, the right column in Fig. 4, the mix of subfigures in Fig. 5, and the validation column in Fig. C2. Also, several axes label relative differences as fractions, but the manuscript text discusses them in terms of percentages. Perhaps it would be easier to interpret if the axes were labelled with percentage values (i.e., -10% instead of -0.1)?

  We agree that some figures include a lot of information. However, with the comprehensive figure captions we provide, we think there is no need to change the figures. It does not seem unusual to us to have several panels with different axes in one illustration.

- **Impact of validation findings:** 3b suggests the model chain has a systematic bias towards underpredicting wet avalanche days at low elevations and overpredicting at high elevations. Are there any known reasons for this and could it be corrected? Furthermore, how does this impact the results? Fig. 3 and 4 show distinct trends in wet avalanche activity by station elevation, but are these results exaggerated due to this bias?

  Thank you for this interesting remark on Fig. 2b (not 3b we suppose). As the underlying model chain (downscaling procedure + SNOWPACK + random forest "blackbox" models) is quite complex, it is hard to tell why wet-snow avalanche days are underpredicted at lower elevations. Interestingly, the validation of the simulated snow height shows a similar systematic bias towards underpredicting snow height, with higher relative differences for lower stations (Appendix Fig. C2 h). Lower snow depths also imply fewer days with snow cover and thus fewer potential wet-snow avalanche days. Yet, this relation is not sufficient to explain observed biases, as the validation does not show the same trend for the dry-snow avalanche days (Fig. 2a). Appendix Figs. C1 8-11 reveal that snow depth at the four lowest stations is particularly underestimated after the peak of snow depth in March / April, which is a relevant time period regarding wet-snow avalanche activity. This underestimation of snow depth at the end of the season can either originate from an underestimation of snowfall (Appendix Figs. C1 15-18), or from an overestimation of melt rates due to differences in the input variables and SNOWPACK settings between the simulations based on the climate scenarios and the simulations based on measurements. For example, incoming shortwave radiation in the climate scenarios is obtained from quantile mapping based on shortwave radiation measurements from a nearby SwissMetNet station, as it is not measured at the IMIS stations. This uncertainty in radiation might also contribute to errors in the simulation of the ablation period and thus influence the parameters used as input for the wet-snow random forest model (e.g. the liquid water content). Yet, finding the ultimate reason for the observed trend in the deviations between simulations driven with climate data versus measurements would require more in-depth station-specific analyses considering all uncertainties inherent in the model chain, and goes beyond the scope of our paper.
  In this context, we have noticed that the manuscript is missing a paragraph on the SNOWPACK settings used in the simulations based on measurements for validation. We apologize for this oversight. We added a corresponding paragraph in Sect. 2.4.

- **Spatial/frequency distribution:** The discussion acknowledges that the model only addressed one component of avalanche risk: the likelihood of triggering. While the importance of destructive size is discussed, the spatial/frequency distribution of avalanches is also a commonly recognized component. The European Avalanche Warning Services defines avalanche danger based on snowpack stability, frequency distribution, and avalanche size. Similarly, the conceptual model of avalanche hazard defines the "likelihood of avalanches" as a function of "sensitivity to triggers" and "spatial distribution." The spatial component of hazard should be acknowledged and discussed. For example, the results show some elevation trends that may relate to frequency distribution, but in general the machine learning models primarily predicted likelihood of triggering. Could the methods be adapted to further examine this, either with spatially distributed snowpack models (e.g., by aspect), or perhaps training machine learning models to predict distribution?

  We agree that the frequency distribution of snow instability is an important element for defining the avalanche danger level in avalanche forecasts. Avalanche forecasts are issued for a certain region. Therefore, it is important to know how many avalanches are to be expected, and of which size. In our study, we simply assess the triggering potential given a certain snow stratigraphy. In other words, there is no spatial component. It is beyond the scope of this study to estimate how the spatial distribution would be affected. In the revised Discussion section (lines 379-383), we now recommend that future research should focus on obtaining projections for different aspects.

- **References:** The manuscript is well cited, but perhaps the reference list is a little long at 5 full pages. Also, I think Verfaillie et al. (2018) is very relevant to this study, and perhaps the recently published review by Eckert et al. (2024).

  Thank you for mentioning these studies which we cite in the updated version of our manuscript. As in our view there are no useless references in our reference list, we have kept all existing references.

**Technical comments**

- Line 110: Appendix B is mentioned before Appendix A. Consider reordering.

  Thank you for noting this. We reordered the appendices.

- Line 178: How was three-day sum of new snow height calculated and was it done the same way for SNOWPACK output and AWS measurements?

  The three-day sum of new snow height provided by SNOWPACK is calculated as the sum of the three consecutive values of the height of new snow within 24 hours. The AWS measurements provide snow depth only and do not measure new snow height. Hence, for this case, SNOWACK was driven with snow depth measurements, and new snow height was calculated taking into account settlement of the snowpack. For the simulations driven with the downscaled climate model output, SNOWPACK was forced with precipitation and the height of new snow was again computed considering settlement. We added this information in the revised version (Sect. 3, lines 223-228).

- Line 181-183: It seems odd to introduce a figure in an appendix before a figure in the main body. I would consider reordering (assuming Fig. 2 is more relevant than Fig. C1).

  We reordered the validation section such that Fig. 2 in the main body is now mentioned before Appendix Fig. C1.

- Line 198: "and with AWS measurements" would better explain relative differences than "or with AWS measurements".

  To enhance clarity, we rewrote the sentence to "To validate the complete model chain, including both dry- and wet-snow avalanche activity models, we calculated the seasonal number of avalanche days (AvDs) using SNOWPACK simulations. These simulations were forced with either downscaled climate models or AWS measurements, and we then computed the relative difference between the resulting seasonal numbers of AvDs." (lines 235-238).

- 3: Do you have any idea why the ensemble of climate models resulted in more spread for wet avalanche days than the dry avalanche days? Was the wet avalanche machine learning model more sensitive to a specific input?

  Thank you for the interesting remark. To answer this question with confidence, a detailed sensitivity analysis comparing both avalanche activity models would be necessary. As this would go beyond the scope of our study, we can only speculate here. The most important meteorological parameter for the dry-snow model is precipitation. For the wet-snow model, on the other hand, radiation and temperature are explicitly included as input features, and precipitation still plays a major role as it drives the length of the snow season and thus the potential length of the wet-snow avalanche season. The dependence of the wet-snow model on the combination of different meteorological parameters can thus be assumed to be stronger compared to the dry-snow model, leading to larger overall variability.

- Line 265: An aside comment… Could you split the dry snow avalanche days by weak layer type to count the number of days with avalanches on persistent versus non-persistent grain types? What other information about the avalanches could be derived from snowpack models (avalanche problems, aspect-trends, etc.)?

  Identifying the weak layer grain type would indeed be an interesting next step. However, this is not as straightforward as one might think, for example, we would have to decide whether to consider all layers exceeding the instability threshold, or just the deepest one or the one with the highest instability value. We therefore did not include this in our analysis. As mentioned above, analyzing trends for different aspects would be an interesting next step. Moreover, analyzing the depths of the weak layers identified by the dry-snow instability model could yield interesting information on changes in potential avalanche size. We added these ideas for future research in the revised Discussion section (lines 365-383).

- Line 297: "and despite" instead of "as despite".

  We intended to express causality with the use of "as" in this case.

**References**

Eckert, N., Corona, C., Giacona, F. et al. Climate change impacts on snow avalanche activity and related risks. Nat. Rev. Earth Environ., 5, 369, https://doi.org/10.1038/s43017-024-00540-2, 2024.

Verfaillie, D., Lafaysse, M., Déqué, M., Eckert, N., Lejeune, Y., and Morin, S.: Multi-component ensembles of future meteorological and natural snow conditions for 1500 m altitude in the Chartreuse mountain range, Northern French Alps, The Cryosphere, 12, 1249–1271, https://doi.org/10.5194/tc-12-1249-2018, 2018.

**References**

Michel, A., Sharma, V., Lehning, M., and Huwald, H.: Climate change scenarios at hourly time-step over Switzerland from an enhanced temporal downscaling approach. Int J Climatol., 41: 3503–3522, https://doi.org/10.1002/joc.7032, 2021

Peleg, N., Molnar, P., Burlando, P. and Fatichi, S.: Exploring stochastic climate uncertainty in space and time using a gridded hourly weather generator. Journal of Hydrology, 571, 627–641. https://doi.org/10.1016/j.jhydrol.2019.02.010, 2019

**Reply to Reviewer #2**

We thank Reviewer #2 for the positive and constructive feedback. Please find below our detailed replies (in blue).

**Specific comments:**

[Line 54 and 76] What constitutes as a "low-elevation" range, and what uncertainties does the "spatial statistical transfer" from lower- to higher-elevation locations introduce that the quantile mapping approach may not address?

Most SwissMetNet stations, for which the CH2018 scenarios are available, are located at elevations below 1500 m a.s.l., so below elevations of typical avalanche starting zones in the Swiss Alps. Nevertheless, some of the stations (e.g. Grand Saint Bernard at 2479 m a.s.l) are located at higher elevations. The reasons why we transferred the projections from SwissMetNet AWS to IMIS AWS are 1) IMIS stations are located at elevations of typical avalanche starting zones, and 2) back-calculating solid precipitation based on snow depth measurements of the IMIS stations gives a more reliable estimate of snowfall amounts, as the SwissMetNet gauges typically experience an undercatch that is particularly strong in winter in case of snowfall. To avoid confusion, we changed the wording to "We apply statistical methods to spatially transfer climate projections from eight members of the CH2018 ensemble to seven automatic weather stations (AWS) located close to typical avalanche starting zones in the Swiss Alps at elevations ranging from 1800 to 2900 m a.s.l." In line 77, we inserted "which are usually located at lower elevations (< 1800 m a.s.l.)". Moreover, we included more information on the quantile mapping approach in Sects. 2.2 and 2.3.

[Line 92 on quantile mapping] I was similarly unsure of the purpose, inner-working, and justification of the multivariate quantile mapping approach. It would be helpful to provide a brief walkthrough and justification of the technique. How exactly are values transformed from CORDEX output to the daily values we use to force SNOWPACK? Does it preserve storm characteristics well enough to allow SNOWPACK to appropriately resolve avalanche-relevant stratigraphy?

We agree that following the multiple steps involved in obtaining climate projections at the IMIS stations, starting from the gridded EURO-CORDEX data, is challenging. We therefore included additional information on this process in the new Sect. 2.2.

[Line 296 on differences across studies] Could the authors comment what they deem the most important factor(s) is/are to reconcile reported differences in warming-induced avalanche trends?

We addressed the discrepancies between studies in the revised discussion (lines 347-353).

**Technical comments:**

[Abstract] Consider providing corresponding numbers of avalanche days in the abstract to give context to the reported percentages.

We inserted this information in the abstract (line 8).

[Line 187 / Figure C1] Please clarify to avoid misinterpreting what values are being compared here for validation. Does this mean, for instance that the 22 values (winters 2001 through 2022) were averaged for 1 Dec, 2 Dec, etc. in the model, and then compared these averages to the observed averages' corresponding day of year? Or, was the entire cold-season averaged, and each season was compared?

We agree that this sentence is somewhat confusing. We first computed the day-of-year average for the variable of interest considering 22 values from the climate simulations spanning the period 2000/2001-2021/2022 and then compared these values to the corresponding values based on the observed data. We have now clarified this in the revised version (lines 247-250).

[Figures 2, 5, and 6] Consider adding elevation values beneath station IDs to help readers interpret results without needing to refer to Fig 1 (or 3). Consider adding a horizontal line at 0 in Fig 2 to emphasize the model's target value.

We agree that adding elevation values below the station IDs might facilitate interpretating results. Yet we found that readability of the figures is limited when adding the elevation values. We therefore included the following information in the respective figure captions: "Stations are arranged in order of increasing elevation, from the lowest (ORT2, 1824 m a.s.l.) to the highest (ARO2, 2847 m a.s.l.)."

We increased the thickness of the line at y = 0 in Fig. 2.

[Line 304] "Enhanced temperatures hinder the formation of weak layers by directly affecting the temperature gradient across the snowpack." Could the authors briefly describe the mechanism for the change in the snowpack's temperature gradient that hinders weak layer formation? Is the gradient itself stronger or more uniform, and why?

We included the explanation in the revised Discussion section (lines 359-364).